# The Influence of Former Process Steps on Changes in Hardness, Lattice and Micro Structure of AISI 4140 Due to Manufacturing Processes

Florian Borchers [1], Brigitte Clausen [2,*], Lisa C. Ehle [3], Marco Eich [1], Jérémy Epp [1], Friedhelm Frerichs [1], Matthias Hettig [2], Andreas Klink [4], Ewald Kohls [1], Yang Lu [5], Heiner Meyer [1], Bob Rommes [4], Sebastian Schneider [4], Rebecca Strunk [1] and Tjarden Zielinski [1]

1   Leibniz-Institute for Materials Engineering, IWT, 28359 Bremen, Germany; borchers@iwt-bremen.de (F.B.); eich@iwt-bremen.de (M.E.); epp@iwt-bremen.de (J.E.); frerichs@iwt-bremen.de (F.F.); kohls@iwt-bremen.de (E.K.); hmeyer@iwt-bremen.de (H.M.); strunk@iwt-bremen.de (R.S.); zielinski@iwt-bremen.de (T.Z.)
2   MAPEX Center for Materials and Processes, University of Bremen, 28359 Bremen, Germany; hettig@iwt-bremen.de
3   Gemeinschaftslabor für Elektronenmikroskopie (GFE), RWTH Aachen University, 52062 Aachen, Germany; ehle@gfe.rwth-aachen.de
4   Laboratory for Machine Tools and Production Engineering (WZL), RWTH Aachen University, 52062 Aachen, Germany; a.klink@wzl.rwth-aachen.de (A.K.); b.rommes@wzl.rwth-aachen.de (B.R.); s.schneider@wzl.rwth-aachen.de (S.S.)
5   BIAS—Bremer Institut für Angewandte Strahltechnik GmbH, 28359 Bremen, Germany; lu@bias.de
*   Correspondence: clausen@iwt-bremen.de

**Abstract:** The surface and subsurface conditions of components are critical for their functional properties. Every manufacturing process modifies the surface condition as a consequence of its mechanical, chemical, and thermal impact or combinations of the three. The depth of the affected zone varies for different machining operations and is related to the process parameters and characteristics. Furthermore, the initial material state has a decisive influence on the modifications that lead to the final surface conditions. With this knowledge, the collaborative research center CRC/Transregio 136 "Process Signatures" started a first joint investigation to analyze the influence of several machining operations on the surface modifications of uniformly premanufactured samples in a broad study. The present paper focusses on four defined process chains which were analyzed in detail regarding the resulting surface conditions as a function of the initial state. Two different workpiece geometries of the same initial material (AISI 4140, 42CrMo4 (1.7225) classified according to DIN EN ISO 683-2) were treated in two different heat treating lines. Samples annealed to a ferritic-perlitic microstructure were additionally deep rolled as starting condition. Quenched and tempered samples were induction hardened before further process application. These two states were then submitted to six different manufacturing processes, i.e., grinding (with mainly mechanical or thermal impact), precision turning (mainly mechanical), laser processing (mainly thermal), electrical discharge machining (EDM, mainly thermal) and electrochemical machining (ECM, (mainly chemical impact). The resulting surface conditions were investigated after each step of the manufacturing chain by specialized analysis techniques regarding residual stresses, microstructure, and hardness distribution. Based on the process knowledge and on the systematic characterizations, the characteristics and depths of the material modifications, as well as their underlying mechanisms and causes, were investigated. Mechanisms occurring within AISI 4140 steel (42CrMo4) due to thermal, mechanical or mixed impacts were identified as work hardening, stress relief, recrystallization, re-hardening and melting, grain growth, and rearrangement of dislocations.

**Keywords:** surface integrity; influencing depth; XRD; SEM; EBSD; hardness; residual stresses; grain size; microstructure; steel AISI 4140 (42CrMo4)

## 1. Introduction

Manufacturing processes change the functional performance of components [1]. Finishing processes, in particular, affect the requested features by changing the workpiece surface layer properties [2,3]. However, the modifications that occur due to machining processes are not always wanted. Processes such as grinding or hard turning affect not only the requested features, but sometimes also generate thermal effects which negatively affect the workpiece surface layer properties, e.g., residual stresses, the microstructure, and the hardness. For example, high residual tensile stresses or large stress gradients may lead to reduced lifecycle performance because of easier crack initiation and propagation. On the other hand, if a surface hardening effect is wanted, a selective heat treatment due to a high thermal impact grinding process can lead to an optimal surface hardness and residual stress state. These examples make clear that it is necessary to acquire the most precise knowledge about the modified microstructures, grain sizes, hardness, and plastic deformations that are generated by each of the processes.

However, up to now it has been very difficult to generate predefined surface layer properties with machining processes [1]. The complexity of the problem increases if several machining operations are directly executed in sequence. To close this knowledge gap a material-oriented view that considers physical and chemical loads which are responsible for the modifications should be applied [4]. The transregional collaborative research center (CRC)/Transregio 136 "Process Signatures" at the Universities of Bremen and Aachen, Germany aims to describe processes by assessing correlations between internal loads, such as temperature and strain fields which can be expressed as process-independent characteristics, and the resulting surface modifications, such as changes in grain size, hardness or residual stresses.

A known process-independent correlation between internal material loads and modifications can allow the inverse problem of the manufacturing technologies to be solved, in order to achieve a predictive process design for targeted workpiece properties [4]. There is already comprehensive research published on the topic of the mentioned inverse problem. For example, in [5] the authors published a detailed description of the mechanisms of the formation of rim zone properties for different manufacturing processes (cutting, waterjet machining, laser beam, electrical discharge machining (EDM) and electrochemical machining (ECM)). In a follow up paper, the authors investigated the functional performance of machined parts. One section described the influence of post machining surface treatments such as shot peening and chemical finishing on the surface properties [6].

Within the CRC, the impact of single process steps on a formerly defined surface and the working mechanisms were thoroughly investigated in a first joint investigation [7]. Since the state of the art for each single machining process considered here and regarding the concept of process signature was already presented in this first study, it will not be repeated here in detail, but can be found in [7].

Further investigations within the CRC [8–14] focus on internal material loads and modifications of single processes. The influence of different initial microstructures on the resulting modification due to single thermal impacts was investigated in [15]. The influence was found to be minor. Only an enhancement in the development of compressive stresses due to an initial FP-state compared to the QT-state could be measured, explained by the larger volume increase in the surface layer.

The question arises regarding what happens if, after an initial machining process, a second but different process is applied. As mentioned above, the complexity of the problem increases rapidly with each manufacturing step. The influence of process chains on the final surface microstructure and functional performance is especially important as the sequence of processing steps can significantly change the resulting properties, as shown by Ehle et al. on the combination of thermal and mechanical treatment of metastable austenitic steel [16].

The present paper tries to approach the problem by investigating a selection of variations of two consecutive manufacturing processes. The selection of processes within this work depicts the possibilities within the CRC projects. For the first step, processes were

chosen that have a larger influence on the modification depth than the second processes, hence a superposition of both processes can be observed. These are induction hardening, as a process with a pure thermal impact, and deep rolling, as a process with pure mechanical impact (Figure 1). Four different defined process chains were considered as initial state for six different manufacturing processes with different proportions of mechanical, thermal, and chemical impacts. While grind-strengthening with main mechanical impact exerts a primarily mechanical influence on the surface, conventional grinding and hard turning induce an additional thermal impact. Exclusive thermal influences were provided by using laser processing. EDM has a main thermal influence, but the surface can, under certain boundary conditions (e.g., rough machining with reduced flushing), additionally be influenced chemically. Since this also happened in this study, the process is not positioned in the triangle corner, but moved slightly to the chemical impact. The impact of ECM on the surface is purely chemical. An overview on the investigated process chains and the process parameters used are given in Section 2.

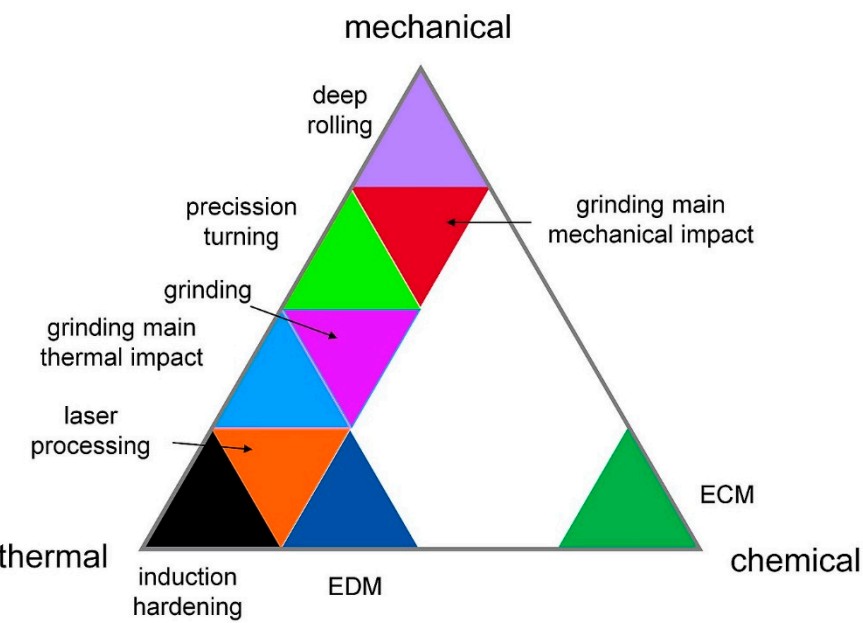

**Figure 1.** The impact of machining operations on the machined surface (the same color code is used in the comparison of the analysis results), as already proposed in [7].

The steel AISI 4140 (42CrMo4) was chosen in the CRC for this thorough investigation since this tempering steel is present in process engineering in different heat treatment states and it allows a wide range of possible property changes due to manufacturing. It is a common steel and therefore purchasable in good reproducibility, though so far all investigations within the CRC could be carried out with one steel batch. As the initial states for this investigation, a defined quenched and tempered and a ferritic-perlitc state, were chosen since these states had been already investigated thoroughly within former investigations in the CRC. Material data for simulations were already collected for these states, so that the results of this study will be further useable for the evaluation of the simulation of comparable process chains.

## 2. Materials and Methods

The steel grade for all samples was the standard AISI 4140 material (42CrMo4, 1.7225, classified according to DIN EN ISO 683-2) which has been used in investigations within this CRC since 2015 (Table 1). Since not every manufacturing process can be applied on the same workpiece geometry, two kinds of samples were analyzed. Processes designed for round samples were performed at cylinders with circular grooves to divide one sample into several sections, as shown in Figure 2a. Processes designed for flat samples were applied

at cuboids (see Figure 2b). The diameter of the cylindrical samples was d = 57.8 mm, the diameter of the grooves was $d_{grooves}$ = 45.8 mm, and the length l = 100 mm for the ferritic-perlitic cylinders. For the cylinders in quenched and tempered condition the length was l = 95 mm. The cuboid samples were 70 mm × 40 mm × 20 mm (l/w/h). First, half of the samples were quenched and tempered to 45 HRC (QT). The heat treatment consisted of a heating of 10 K/min up to 850 °C and holding for 2 h at 850 °C followed by oil quenching to 60 °C and a tempering at 400 °C for 4 h. The other half was heat-treated to a ferritic-perlitic microstructure by normalizing with a final hardness of 211 HV 1 (FP). The mechanical properties in this initial state determined by tensile tests are shown in Table 2. Subsequently, the hardened samples were induction hardened and the ferritic-perlitic samples were deep rolled. The respective process parameters for these processes are provided in detail in the following section. After these treatments, the circular grooves, separating the surface into sections, were turned into the cylinders and the samples were provided for further processing. The geometry of the grooves was the same for both material conditions, though the FP cylinders were longer and were equipped with a holding pin (as can be seen on the right side of Figure 3a, which was needed for clamping in the deep rolling process.

The initial ferritic-perlitic FP state, as measured by X-ray diffraction, has less than 40 MPa of residual stresses remaining through the whole volume, and reveals a peak width FWHM of around 0.56° and a Martens hardness of HM 2250 N/mm². The second condition with a quenched and tempered QT state of the same samples geometry also shows less than 40 MPa residual stresses in the outer layer (0–2 mm, core excluded), a FWHM value of 3.5°, and a Martens hardness of HM 4250 N/mm².

**Table 1.** Chemical composition of the used steel melt.

| Notation | C | Cr | Mn | P | S | Si | Mo | Ni | Al | Cu |
|---|---|---|---|---|---|---|---|---|---|---|
| Unit | % | % | % | % | % | % | % | % | % | % |
| AISI 4140/42CrMo4 | 0.448 | 1.09 | 0.735 | 0.012 | 0.002 | 0.264 | 0.244 | 0.200 | 0.018 | 0.065 |
| DIN EN ISO 683-2 | 0.38–0.45 | 0.90–1.20 | 0.60–0.90 | max. 0.025 | max. 0.010 | max. 0.40 | 0.15–1.30 | – | – | – |

**Table 2.** Mechanical properties of the material conditions of the heat treatment states used in this investigation.

| Notation | Initial Material Condition | |
|---|---|---|
| | **FP** | **QT** |
| yield strength (YS) | 380 MPa (YSFP) | 1420 MPa (YSQT) |
| ultimate tensile strength (UTS) | 730 MPa (UTSFP) | 1560 MPa (UTSQT) |

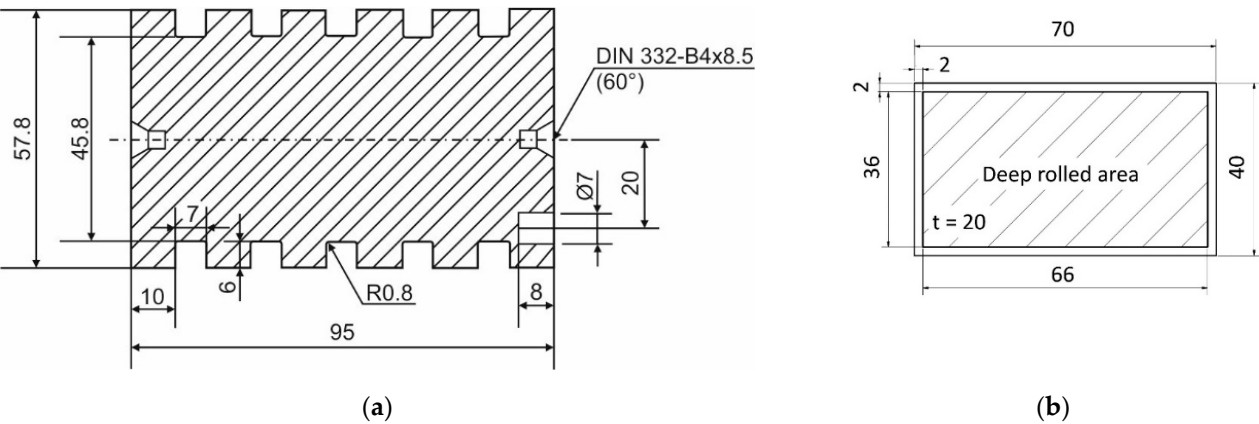

(**a**)　　　　　　　　　　　　　　　　　　　　　　　　　　(**b**)

**Figure 2.** Drawings of the used samples: (**a**) Cylindrical sample with different sections separated by grooves (exemplarily shown for the induction hardened state) and (**b**) cuboid sample geometry (exemplarily shown for the deep rolled state), with dimensions in mm.

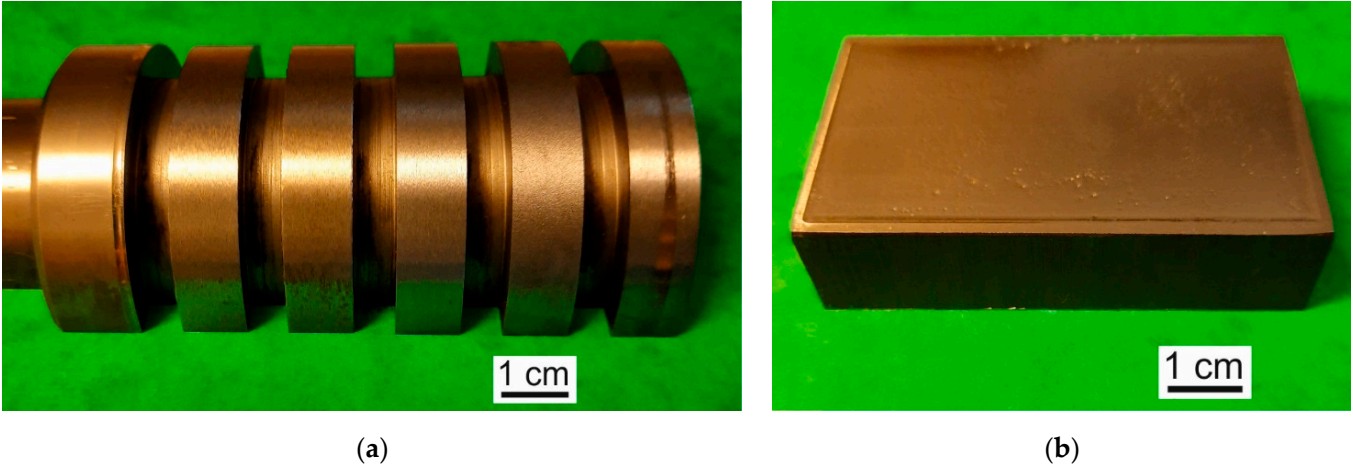

(**a**)                                                                                              (**b**)

**Figure 3.** Photos of the used samples: (**a**) example of a cylindrical sample in FP state after machining and being cut for further investigation and (**b**) example of cuboid in FP state with deep rolled surface.

The samples were delivered to the different machining processes. After machining, the samples were submitted to the different analyzing techniques. For each machining process, two sets of manufacturing parameters were applied, in order to induce different process impacts. The following chapters provide the machining conditions and parameters in detail, while the overview over the complete experimental plan including the different processes and the performed characterizations is presented in Figure 4.

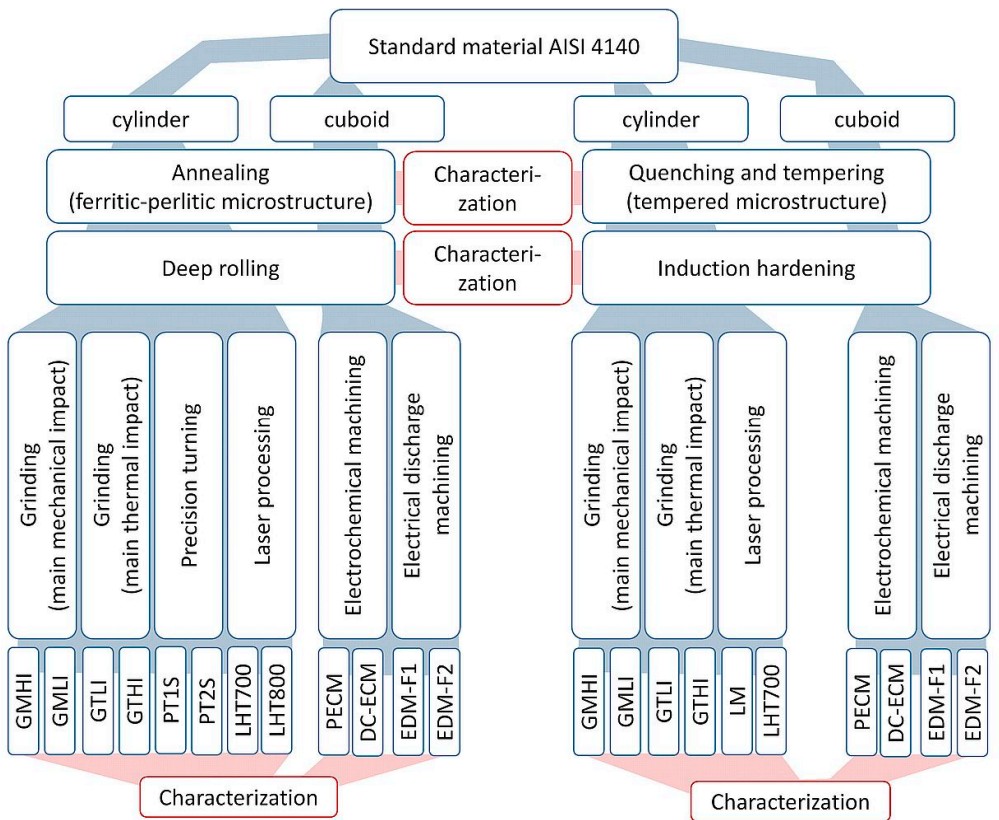

**Figure 4.** Overview of the experimental plan.

## 2.1. Deep Rolling

Deep rolling was conducted on a CNC turning machine for the cylindrical specimens and on a conventional 4-axis CNC manufacturing center for the cuboid specimen. To

conduct the process, a hydrostatic deep rolling tool was used. Hydrostatic deep rolling is a force-controlled process; the deep rolling normal force is not generated by the man­ufacturing center, but by an external hydraulic unit. A fluid coolant which transmits the necessary force is used and transferred to the tool with pressures up to 400 bar. The tools are equipped with a lift up to 5 mm to compensate unevenness and to ensure constant contact to the workpiece. As a result, the applied force remains constant, even if slight height differences occur during the process. A single-pass deep rolling operation was conducted on the surface using the parameters given in Table 3. The parameters were chosen with the aim of introducing high compression residual stresses while creating a smooth surface. The deep rolling velocity $v_{dr}$ is the only changing process parameter between the cylindrical and cuboid samples. The difference is based on machine dynamic limitations and has no significant influence on the resulting residual stresses according to Röttger [17]. The deep rolling force is assumed to be a static in the used velocity range [17].

**Table 3.** Parameters for deep rolling operation.

| Ball Diameter $d_b$ [mm] | Deep Rolling Pressure $p_{dr}$ [bar] | Deep Rolling Force $F_{dr}$ [N] | Feed $f_{dr}$ [mm/rev] | Deep Rolling Velocity $v_{dr}$ [mm/min] |
|---|---|---|---|---|
| 13 | 159 | 2107 | 0.04 | 100 (cylinders) 0.75 (cuboids) |

### 2.2. Induction Hardening

The induction heating unit is a universal hardening machine of type VL 1000 SINAC 200/300 S MFC manufactured by EFD Induction (Skien, Norway). The plant works in a middle frequency range from 10 to 15 kHz and in a high frequency range between 100 and 200 kHz. A rectangular inductor coil for the cuboids was used in combination with a magnetic flux controller Ferrotron 559. This was necessary to concentrate the magnetic flux density and increase the generated heat density within the sample. The cuboids were heat-treated within the infield with a middle frequency of 10.8 kHz. The coil moved with a feeding velocity $v_{ft}$ of 900 mm/min in a distance of 2 mm along the sample surface, which was cooled down immediately after the heating with a polymer–water solution with a volume flow of 23 L/min.

The cylinders were heat-treated with an inductor coil consisting of one round coil with an inner diameter of 62 mm. A magnetic flux controller was not used. The cylinders were heat-treated within the infield of the inductor coil with a high frequency of 180 kHz. The directly following quenching was carried out again with a polymer–water solution, with a volume rate of 23 L/min. Table 4 summarizes the main parameters of the induction heating processes.

**Table 4.** Parameters for inductive heat treatment operation.

| Workpiece | Inductor Size $S$ [mm] | Feed $v_{ft}$ [mm/s] | Maximal Generator Current $I_{max}$ [A] | Frequency $f$ [kHz] |
|---|---|---|---|---|
| cuboid | 50 (length) | 900 | 478 | 10.8 |
| cylinder | 62 (inner dia.) | 500 | 194 | 180 |

### 2.3. Grinding

The grinding experiments were performed on a Studer S41 high precision cylindrical grinding machine on the cylindrical workpieces. The process was carried out as external cylindrical grinding with a constant radial feed rate $v_{fr}$ and only one process stage for each setup. Additionally, a spark out with no radial infeed was applied for $t_s = 1$ s. The used grinding wheels were both vitrified bond corundum wheels with different specifications, as shown in Table 5. A dressing procedure was performed before each experiment to generate similar tool surface properties. For the dressing procedure, a form roller was used. The

dressing speed ratio q = 0.6 and the overlapping rate in dressing $U_d$ = 3 were kept constant. As a metalworking fluid, a universal grinding oil was used. The cooling conditions were kept constant over all experiments. No further preliminary processing was applied to the workpieces before the described grinding processes. The resulting changes, e.g., in microstructure and surface integrity are therefore caused by the modifications due to the grinding process. The workpieces were mounted in the grinding machine between centers and adjusted to obtain a possibly precise (low) rotational deviation.

**Table 5.** Parameters and set up for grinding operation.

| Level | Grinding Wheel | Cutting Speed $v_c$ (m/s) | Radial Feed Rate $v_{fr}$ (mm/min) | Depth of Cut $a_e$ (mm) | Direction | Depth of Dressing Cut $a_{e,d}$ (mm) |
|-------|----------------|-------------------------|-----------------------------------|------------------------|-----------|-------------------------------------|
| GMHI | A60P5AV | 1.36 | 0.20 | 0.1 | Down-grinding | 3 × 0.02 |
| GMLI | A60P5AV | 14.76 | 0.20 | 0.1 | Down-grinding | 3 × 0.02 |
| GTLI | A80CC5V | 35.18 | 2.45 | 0.3 | Up-grinding | 5 × 0.02 |
| GTHI | A80CC5V | 35.18 | 4.90 | 0.3 | Up-grinding | 5 × 0.02 |

The process parameters were selected to cover a range of process impacts and combinations, from the rather mechanical main impact over thermo-mechanical impacts up to the rather dominating thermal impact of the grinding process on the workpiece. This was adapted from the concept to represent different possible modifications and mechanisms caused by grinding processes depending on the setup and the boundary conditions. Table 5 shows the utilized process parameters. Setup 1 was adjusted with a rather mechanical main impact (grinding mechanical high intensity (GMHI)).

Up to setup 4, the thermal impact should become more relevant or dominating, respectively (grinding thermal high intensity (GTHI)).

## 2.4. Precision Turning

The precision machining experiments were performed at the cylindrical workpieces on a precision lathe GoFuture B2 (Carl Benzinger GmbH, Pforzheim, Germany) and carried out as a turning process. A new cubic boron nitride tool (S274 TH35, Hartmetall-Werkzeugfabrik Paul Horn GmbH, Tübingen, Germany) was used for all experiments. The cutting speed and feed were held constant during the processes with $v_c$ = 100 m/min and f = 18 μm/rev. During the cutting experiments, the process forces in terms of cutting force $F_c$ and thrust force $F_t$ were measured with a multicomponent dynamometer MiniDyn 9119AA1 (Kistler Instrumente AG, Winterthur, Switzerland) with a sampling rate of 1kHz. A variation in process parameters was realized with a multistage process and a summarized depth of cut $a_p$ = 50 μm. In one experiment, the specimen was machined in one stage and $a_p$ = 50 μm (1 stage precision-turning (PT1S)), while in the other experiment, in the first stage $a_{p,1}$ = 30 μm, and afterwards $a_{p,2}$ = 20 μm, were applied (2 stage precision turning (PT2S)). To measure $a_p$ and account for any misalignment between tool and workpiece, a part of the cylinder was left for characterization. The parameters are summarized in Table 6.

**Table 6.** Parameters for precision turning operation.

| Level | Cutting Speed $v_c$ [m/min] | Feed f [μm/rev] | 1. Depth of Cut $a_{p,1}$ [μm] | 2. Depth of Cut $a_{p,2}$ [μm] |
|-------|----------------------------|-----------------|-------------------------------|-------------------------------|
| PT1S | 100 | 18 | 50 | 0 |
| PT2S | 100 | 18 | 30 | 20 |

## 2.5. Laser Processing

The experimental setups for laser processing consist of a fiber laser (JK400FL, GSI Lumonics, Warwickshire, UK) with a wavelength of 1070 nm and a disc laser (TruDisk 12002, Trumpf, Ditzingen, Germany) with a wavelength of 1030 nm. Both laser systems work in continuous-wave (cw) mode. The fiber laser was used for laser remelting and the disc laser was applied for laser heat treatment of cylinder-shaped AISI 4140, respectively, as shown in Figure 5. The used laser processes are summarized in Tables 7 and 8. For laser melting (LM), the laser was guided into a scan head (Superscan III-15, Raylase, Wessling, Germany) and focused with a telecentric F-Theta optic (S4LFT3162, Sill Optics, Wendelstein, Germany). The laser scanned bidirectionally in the direction of the axis of the probe. Meanwhile the probe was rotated by a rotation motor (URB100CC, Newport, Darmstadt, Germany) so that the whole surface was processed. For the laser heat treatment (LHT), the laser was guided into a laser head (BEO D D70 90°, Trumpf, Ditzingen, Germany) and a shaped rectangular beam was focused on the surface statically. Then, the cylinder was rotated by a rotation motor (RT1A, Föhrenbach, Löffingen-Unadingen, Germany) to perform the laser hardening. During the laser heat treating, the laser irradiated area was analyzed by a two-color pyrometer (IGAR12-LO-MB22, Lumasense, Frankfurt, Germany) to record the transient temperature course.

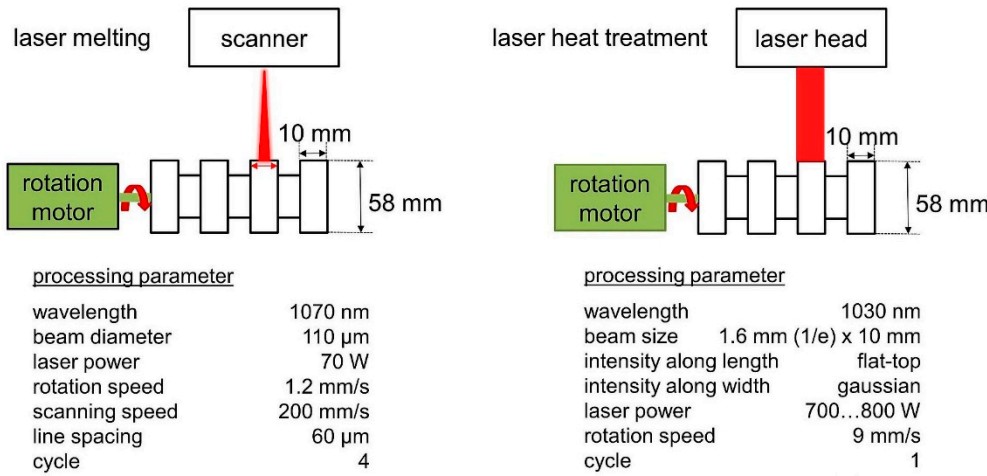

**Figure 5.** Schematic illustration of the experimental setup, machining setup and parameters of laser remelting and laser hardening processing.

**Table 7.** Parameters for cw laser melting operation.

| Level | Input State | 1st Process | 2nd Process | Laser Power [W] | Focus Diam. [μm] | Scan Velocity [mm/s] | Line Spacing [μm] | Rot. Speed [mm/s] | Overrun Cycles [–] |
|---|---|---|---|---|---|---|---|---|---|
| LM | QT | Inductive hardening | Laser melting | 70 | 110 | 200 | 60 | 1.2 | 4 |

**Table 8.** Parameters for cw laser heat treatment operation.

| Level | Initial State | 1st Process | 2nd Process | Laser Power [W] | Rot. Speed [mm/s] | Beam Size [mm²] | Overrun Cycles [–] | Surface Temperatur [°C] |
|---|---|---|---|---|---|---|---|---|
| LHT700 | QT | Inductive hardening | Laser hardening | 700 | 9 | 10 × 1.5 | 1 | 1250 |
| LHT700 | FP | Deep rolling | Laser hardening | 700 | 9 | 10 × 1.5 | 1 | 940 |
| LHT800 | FP | Deep rolling | Laser hardening | 800 | 9 | 10 × 1.5 | 1 | 1240 |

### 2.6. Electro-Discharge Machining (EDM)

The experiments for the EDM process were conducted on a GFMS FORM 2000 VHP sinking EDM machine tool. A hydrocarbon-based dielectric fluid was used (Ionoplus IME-MH) and a tool electrode made of fine graphite (ELLOR® +50) with a cross-section area of $25 \times 25$ mm$^2$ was used at the cuboid samples. The applied parameter sets can be described as finishing parameters, with a removal depth of $z_1$ = 250 μm for EDM-F1 and a removal depth of $z_2$ = 1000 μm for EDM-F2. An overview is listed in Table 9.

**Table 9.** Parameters for EDM operation.

| Level | Open Circuit Voltage $\hat{u}_i$ [V] | Discharge Current $i_e$ [A] | Discharge Duration $t_e$ [μs] | Pulse Interval Time $t_0$ [μs] | Machining Depth $z$ [μm] |
|---|---|---|---|---|---|
| EDM-F1 | 100 | 8 | 14.1 | 28.9 | 250 |
| EDM-F2 | 100 | 3.4 | 8.4 | 13.1 | 1000 |

### 2.7. Electro-Chemical Machining (ECM)

The experiments for the ECM process were conducted on an EMAG PTS1500 machine tool (EMAG GmbH & Co. KG, Salach, Germany). For both initial material states, cuboid specimens were shortened to an edge length of 12 mm by using Wire-EDM cutting. This shape is optimized for the usage in a special PECM-capable fixture, which provides optimal flushing conditions even for such small-scale ECM experiments. For these investigations, two sets of parameters were chosen: The first set corresponds to an ECM roughing operation, which means the stationary process variant is applied. For this operation a voltage of 15 V and a feed rate of 0.5 mm/min were set to machine a depth of 0.25 mm of the pre-treated surfaces. The electrolyte flushing (sodium nitrate) occurred at a pressure of 2 bar, a temperature of 22.5 °C and a corresponding electric conductivity of 103 mS/cm. The second set of parameters corresponds to a finishing operation and therefore uses the pulsed variant of ECM, called PECM. For this, a voltage of 25 V was applied during short electric pulses, which had a length of 1.1 ms and a pause duration of 1 ms. The pulses were only applied near the bottom-dead-center of the mechanical tool-oscillation, which occurred at a rate of 50 Hz. In this case, the feed-rate was set to 0.075 mm/min in order to once again remove 0.25 mm from the pre-treated surfaces.

Apart from the inlet pressure, which was set to 4 bar, the electrolyte conditions were identical to the stationary parameter-set. An overview is listed in Table 10.

**Table 10.** Parameters for ECM operation.

| Level | Voltage $U$ [V] | Feed Rate $f$ [mm/min] | Pulse Duration $t_p$ [μs] | Pause Duration $t_0$ [μs] | Inlet Pressure $P$ [bar] |
|---|---|---|---|---|---|
| DC-ECM | 15 | 0.5 | – | – | 2 |
| PECM | 25 | 0.075 | 1000 | 1000 | 4 |

### 2.8. Analyzing Techniques

To gain information about the characteristics and the depth of the material modifications, different analyzing techniques were used. Martens hardness measurements were used to determine the hardness changes caused by the processes as a function of depth. In contrast to conventional hardness measurements, Martens hardness involves very small forces which are suitable for analyzing small distances from the surface at prepared cross sections.

Electron microscopy, with its manifold analysis possibilities, provides information concerning microstructural changes.

Based on the measurements of the crystalline phases and changes in the lattice parameters, X-ray diffraction analyses were applied to determine the residual stress state and the

peak width at the processed surface. Additionally, in-depth analysis by electrochemical layer removal were used to determine residual stress and peak broadening depth profiles.

To visualize the resulting changes in the measured depth profiles, the obtained values after processing are plotted as a function of the total distance from the initial surface, which means that the layer thickness removed by the considered process was used as starting point. This was applied to the hardness measurements, the microstructure analysis by electron microscopy, and the X-ray diffraction results. The procedure is exemplarily shown in Figure 6 for a residual stress depth profile. The initial cylindrical sample was annealed to a ferritic-perlitic state and deep rolled. One part of the cylinder was measured in the initial deep rolled state (square symbols, displayed for two single cylinders). A second region of the cylinder was measured after additional precision turning, removing 50 μm from the initial surface (diamond-shaped symbol). The first measured value was therefore plotted at 50 μm depth on the X-axis, representing the distance from the initial surface. This measure enabled a direct comparison of the material condition at identical initial depth positions. It became clear that in a further 50 μm distance from the new surface, the effect of precision turning had subsided.

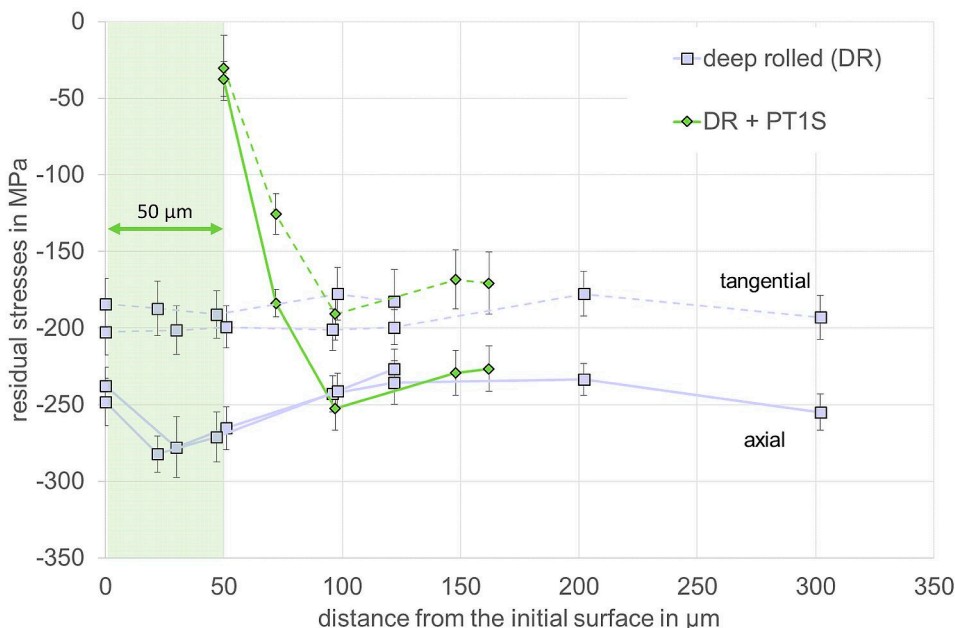

**Figure 6.** Schematic illustration of the mutually shifted depth profiles.

### 2.8.1. Hardness Measurements

The Vickers hardness measurements were carried out according to DIN EN ISO 6507 [18]. The Vickers hardness was measured at previously prepared cross sections only in the initial quenched, tempered and induction hardened, annealed and deep rolled states, respectively, to visualize the hardness depth profiles after these treatments, which represent the respective initial conditions.

Universal micro hardness (UMH) [19] was measured at previously prepared cross section of the samples with a Vickers indenter using a Fischerscope H100C (Helmut Fischer GmbH, Sindelfingen, Germany). The testing force was set to 10 mN and held for 10 s. The Martens hardness was calculated based upon the force depth course, considering the elastic and plastic deformation beneath the indenter, which qualifies it to provide information about the hardness with very small indentations. In order to place the indentations as close as possible to the processed surface on the cross sections, three depth profiles were measured running obliquely into the depth of the material, as shown in Figure 7. The indentations up to a distance of 5 μm to the surface do not conform to the standard, and can therefore only be used as indicators of the achieved modification of the material.

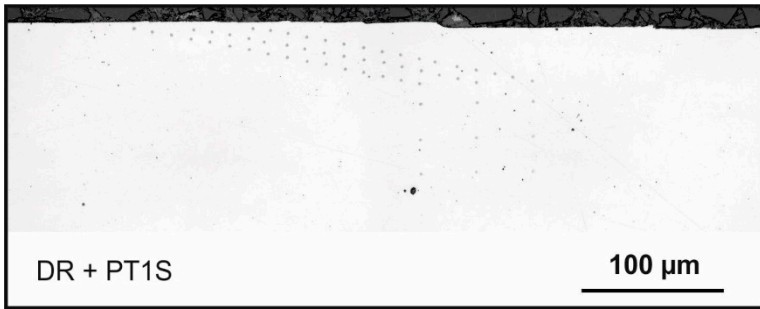

**Figure 7.** Example for the obliquely placed indentations to measure the hardness of the near surface area.

### 2.8.2. Scanning Electron Microscopy (SEM), Energy-Dispersive X-Ray Spectroscopy (EDS) and Electron Backscatter Diffraction (EBSD) Measurements

Since light microscopic investigations showed no pronounced difference between the machined samples, overviews of the machined surfaces were obtained on etched cross sections using a scanning electron microscope. Specimens were embedded in resin, ground, polished and etched with 3% nitric acid ($HNO_3$) in alcohol. Since the resin is nonconductive, the samples were sputtered with gold after etching. The overviews were taken with a Vega II XLH electron scanning microscope (Tescan GmbH, Dortmund, Germany) at the Leibniz-Institute for Materials Engineering-IWT Bremen.

For further electron microscopical investigations at the central facility for electron microscopy (GFE), Aachen, another specimen was nickel plated and embedded in a conductive phenol resin (ATM GmbH, Mammelzen, Germany). The cross-section was ground with SiC paper and then polished with a diamond suspension of 3 μm, 1 μm and 0.5 μm. The final polishing was performed with a colloidal 0.05 μm $SiO_2$ suspension.

Electron backscatter diffraction (EBSD) measurements were acquired using a JSM7000F scanning electron microscope (JEOL, Tokyo, Japan) equipped with a Hikari EBSD detector (Ametek EDAX, Weiterstadt, Germany). The data collection and analysis were performed using the OIM Data Collection and OIM Analysis Software from EDAX-TSL. The step size for EBSD measurements was set to 70 nm. The acceleration voltage for EBSD measurements was 15 keV.

### 2.8.3. X-ray Diffraction

To measure the residual stress state and changes in the crystalline structure of the modified sample surfaces, the X-ray diffraction technique was employed. The depth profiles were obtained using a stepwise electropolishing in phosphoric/sulfuric acid electrolyte and successive measurement in a standard calibrated diffractometer using a Cr-Kα radiation source and a gas-filled line detector at Leibniz Institute for Materials Engineering-IWT Bremen. The α-{211} plane and its diffraction peak were selected for measurement and analyzed following the standard $\sin^2\Psi$ technique [20], obtaining both stress values and peak width (FWHM). Measurements were always performed in two directions (e.g., axial/tangential or longitudinal/transversal to the sample geometry). Constants used for the calculation were the elastic modulus of the α-{211} peak with E = 220 GPa and the Poisson ratio with ν = 0.28. All measurements were carried out following a fixed template as detailed in Table 11. The full width at half maximum value (FWHM) of the diffraction peak provides information about material changes such as work hardening, recrystallization and dislocation annihilation or ordering [21]. Broadening of the diffraction peak is typically associated with a decrease in domain size and increasing microstrains which may be connected to a decreasing grain size and an increasing dislocation density [8]. Effects of the layer removal on the residual stress values for the depth profiles were intentionally not corrected.

**Table 11.** Parameters for the measurement of residual stresses by X-ray diffraction.

| Parameter Abbr. [unit] | Primary Beam Diameter db [mm] | Lattice Plane | Tube Voltage U [kV] | Tube Current I [mA] | Ψ Angles 11 between | Step [°2θ] | Range [°2θ] |
|---|---|---|---|---|---|---|---|
| value | 2 | α{211} | 33 | 40 | −45° to +45° | 0.1 | 147 to 163 |

## 3. Results

### 3.1. Comparison of Hardness Measurement Investigations Results

The Vickers hardness and the Martens hardness of the cylinders QT + IH are shown in Figure 8a,b. The Vickers hardness up to a depth of about 800 μm is about 730 HV1, which equals about 6500 N/mm$^2$ Martens hardness for the inductively hardened surface layer. Figure 8c,d compares the depth profiles of two sets of parameters each for different process chains together with the hardness profiles evolving from inductive hardening. The grinding process with high mechanical main impact (GMHI) exhibits only a slight influence on the hardness (see red line with filled circles in Figure 8c). Grinding with high thermal impact reduces the hardness in the surface region, as can be seen in the blue line with blank circles in Figure 8c. In the region of the measurements the values start at strongly reduced hardness and increase with increasing depth towards the initial state with about 200 μm. The laser melting process impacts the hardness in a similar way as the GTHI process as the hardness is on a similar level with this process (see orange line with blank rhombuses in Figure 8c).

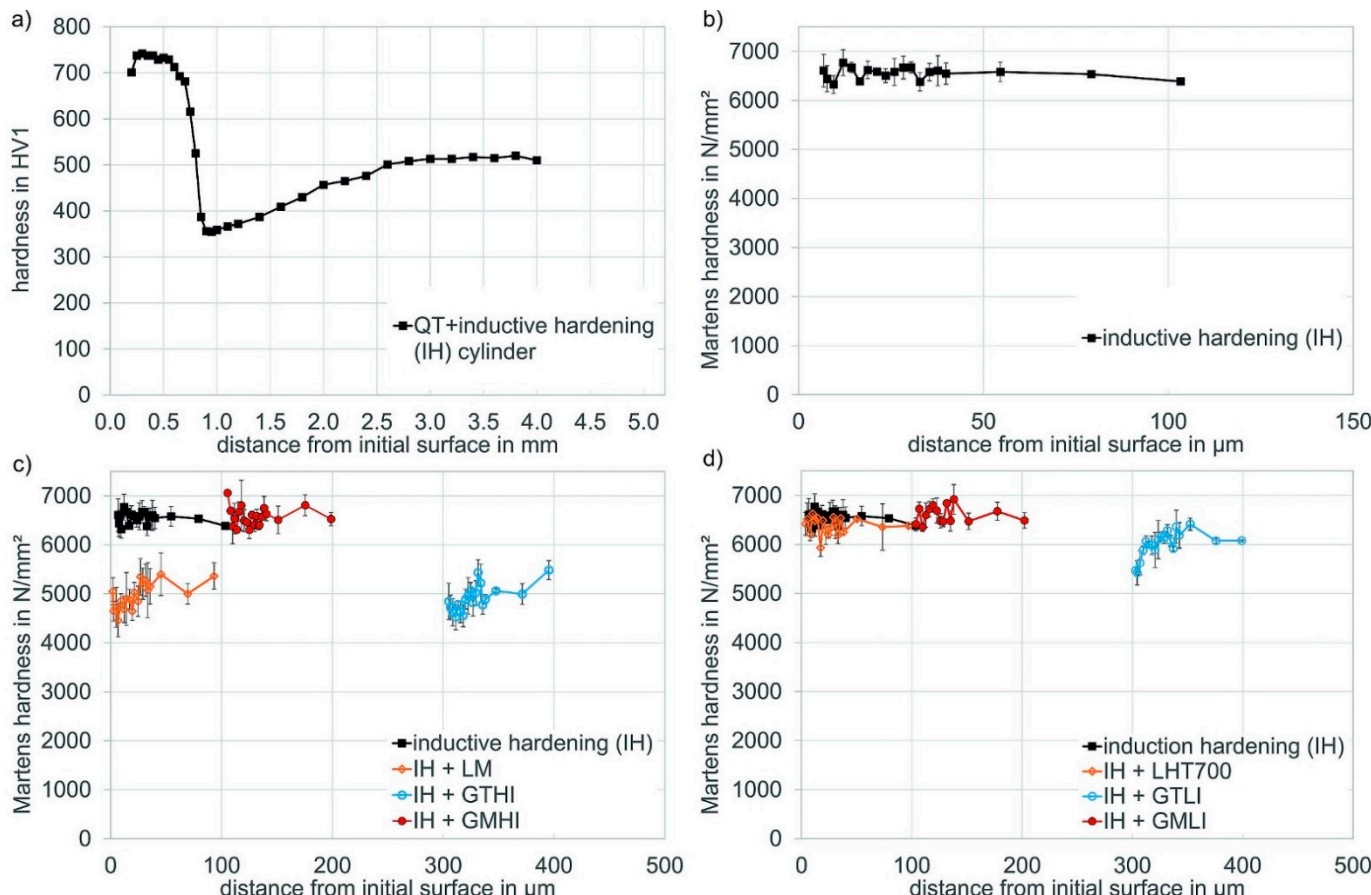

**Figure 8.** Hardness depth profiles of cylindrical specimens in QT-condition applied with experimental process chains consisting of: (**a**) QT + IH Vickers hardness HV1 and (**b**) Martens hardness HM, (**c**,**d**) Martens hardness depth profiles of two sets of parameter each for three process chains.

The grinding process with low mechanical main impact GMLI (red line with filled circles in Figure 8d) and the LHT700 process do not lead to a strong change in the hardness. It stays on the same level as the hardness of the IH specimen (black line with filled squares). The hardness is lowered in a depth of about 50 μm due to the grinding process with low thermal impact (GTLI).

EDM and ECM processes carried out on induction hardened cuboids showed no impact on the hardness apart from the removal. Therefore these results are not shown here.

The hardness of the FP + DR cylinders is depicted in Figure 9a,b (Vickers hardness, Martens hardness, respectively); see lilac lines with filled squares. Deep rolling induces a slight increase in the hardness in the FP initial material state towards the treated surface, due to cold working. The value of the Martens hardness is about 3000 N/mm$^2$ in the first 80 μm. Again, process chains with two sets of parameter each for different process chains are compared in Figure 9c,d. The laser process with both parameter sets LHT700 and LHT800 has the most significant effect on the hardness. They result in an increase in the hardness of different characteristics in the near surface region (see orange line with blank diamonds in Figure 9c,d). Precision turning shows no effect on the hardness, as the value fits well with the values of the DR specimen (see light green line with filled diamonds in Figure 9c,d). The grinding processes with mechanical main impact increase the hardness within the first 20 μm of the surface (see red line with filled circles Figure 9c,d). As for the grinding process with thermal main impact, the hardness of the FP + DR material increases in the first 50 μm of the depth profile. The higher thermal impact (see Figure 9c) shows a more pronounced influence than the low thermal impact (see blue line with blank circles in Figure 8d).

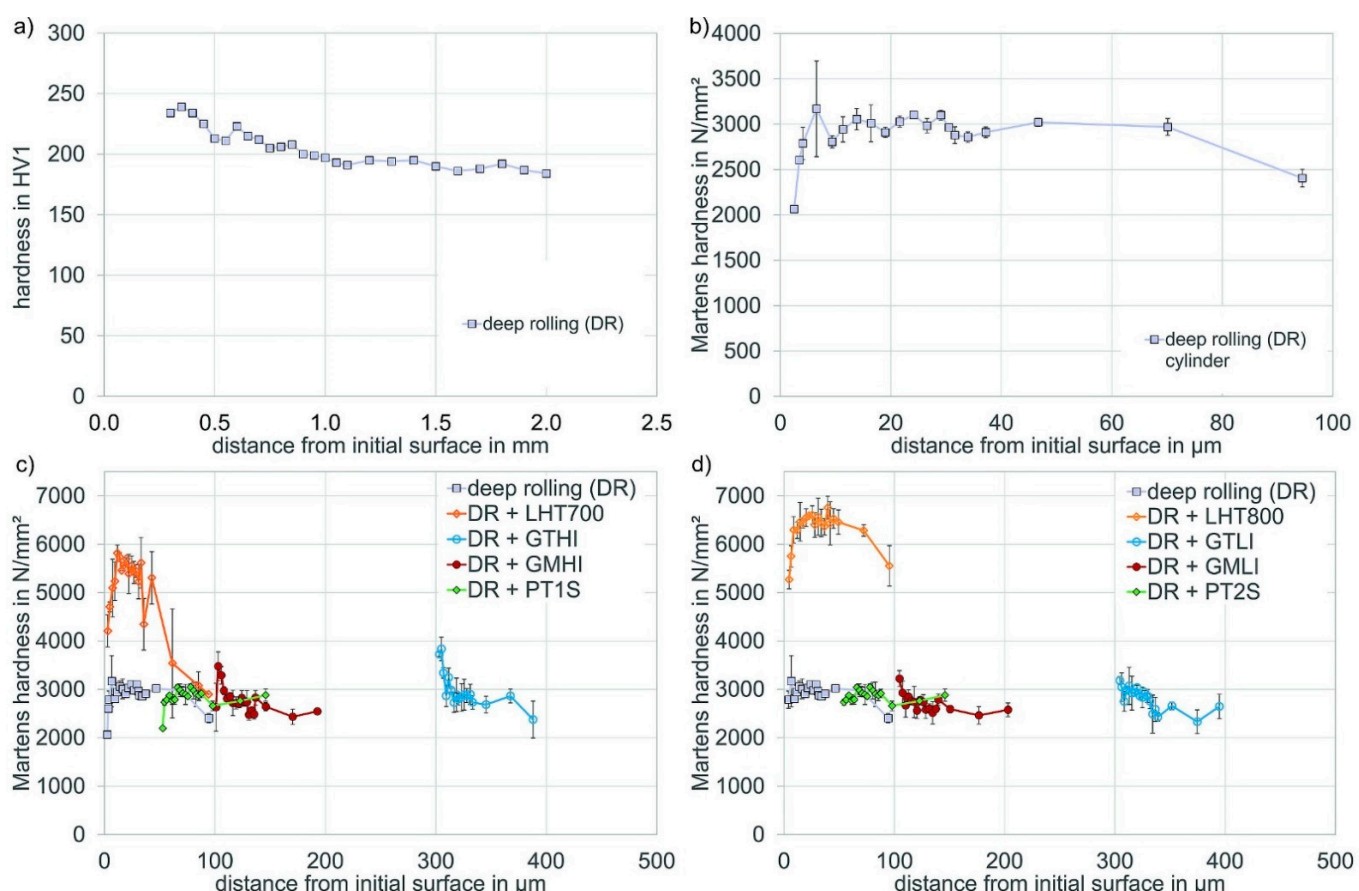

**Figure 9.** Hardness depth profile of cylindrical specimens in FP-condition applied with experimental process chains consisting of deep rolling (DR) + x: (**a**) FP + DR Vickers hardness depth profile, (**b**) FP + DR Martens hardness depth profile, (**c,d**) Martens hardness depth profiles of two sets of parameter each for different process chains.

The processes EDM and ECM carried out on deep rolled samples showed no effect on the Martens hardness apart from the removal. For this reason, the hardness measurement results on cuboids are not represented in this paper.

### 3.2. Comparison of Results Achieved with Electron Microscopy

3.2.1. Overview of the Outer Layers

The effects on the microstructure of the process chains are presented in the following micrographs taken at the cross sections. The microstructure of an inductive hardened QT specimen is depicted in Figure 10. A typical microstructure of a hardened 42CrMo4 steel is visible. The very narrow edge, which appears bright white, is an artefact and can be traced back to the charging in the SEM.

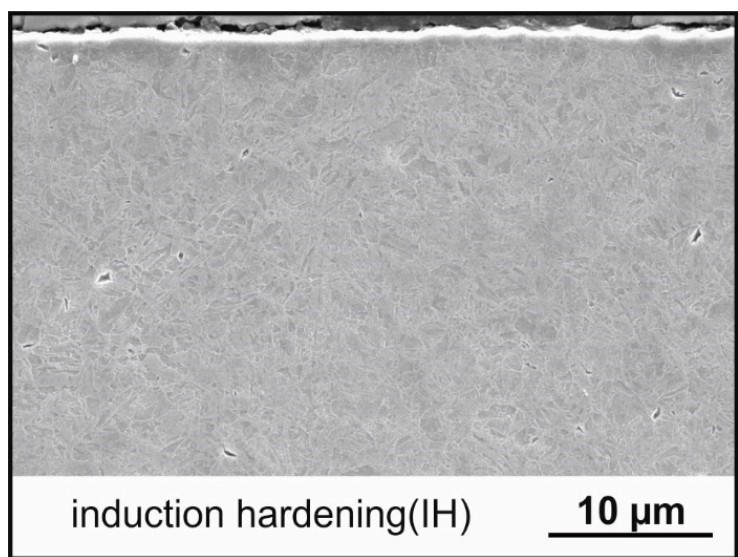

**Figure 10.** SE-micrograph of the initial QT + IH material condition (cylinder).

The influence of the grinding processes on the microstructure is shown in Figure 11. The micrographs are sorted from a high mechanical intensity (Figure 11a) to a high thermal intensity (Figure 11d), with the lower process intensities in between. The GMHI results in a strongly deformed near surface region of about 3 to 4 μm (see Figure 11a). However, the GMLI process has a less pronounced effect on the near surface microstructure and the effect of the grinding direction to the left is barely visible. Furthermore, a deformed near surface region cannot be identified (see Figure 11b). Grinding with low thermal intensity shows a small fringe in the surface area, where the microstructure appears homogeneous and deformation in grinding direction cannot be detected (see Figure 11c). A clearer fringe can be detected with the GTHI process (see Figure 11d). The microstructure underneath the GT specimen appears finer than that of the GM specimens.

The microstructure of the initial FP + DR microstructure is shown in Figure 12. The effects of the deep rolling process are not visible in the SE-micrographs. Only the initially present FP microstructure consisting of ferrite and perlite grains is visible.

The effects of GMHI on the near surface region are clearly visible (Figure 13a). The near surface region is strongly deformed in the grinding direction. The deformed cementite lamellae are clear indicators for the mechanical deformation. A narrow fringe of in grinding direction deformed cementite lamellae results from the GMLI process (see Figure 13b).

With the GTLI process, the deformation in grinding direction is visible as a narrow fringe of cementite lamellae at the surface, see Figure 13c. GTHI results in a surface zone of two to three micrometers deformed in grinding direction, consisting of former narrow cementite lamellae that appear to have joined, see Figure 13d.

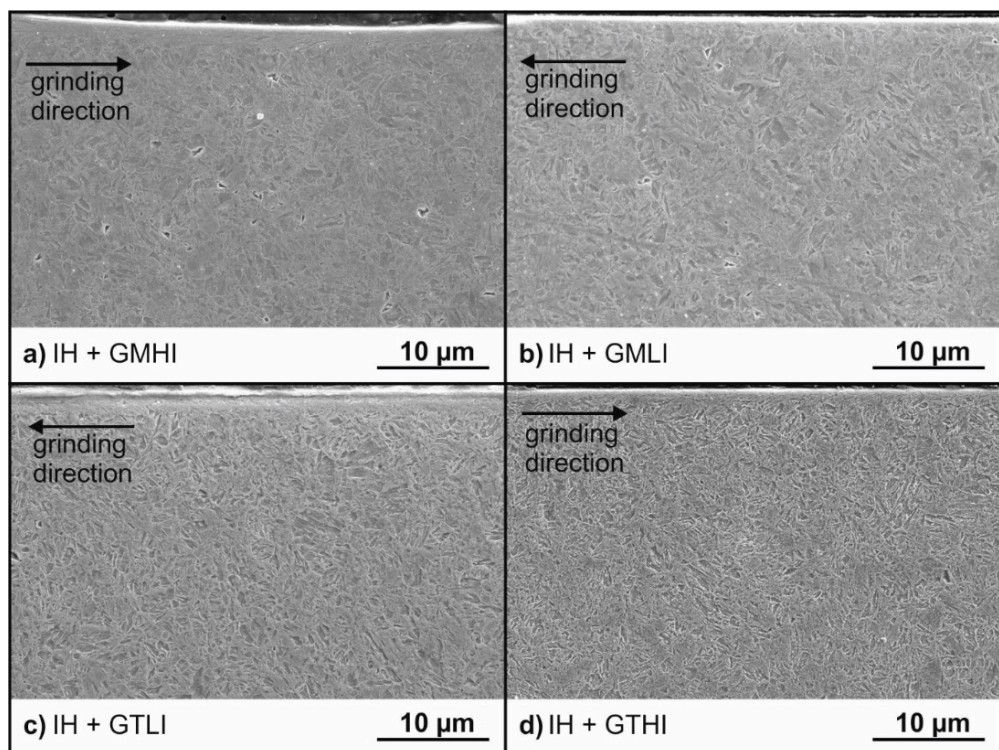

**Figure 11.** Near surface microstructure of process chains consisting of QT + IH +: (**a**) GMHI, (**b**) GMLI, (**c**) GTLI, and (**d**) GTHI (cylinder).

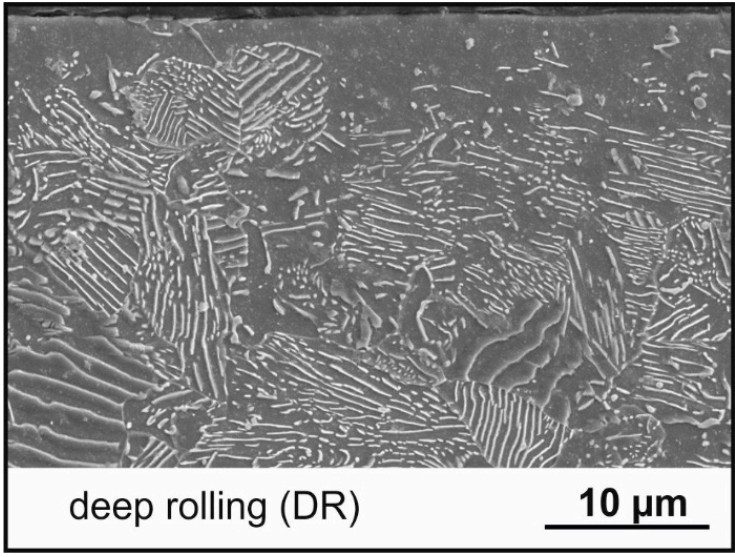

**Figure 12.** Ferritic-perlitic microstructure after deep rolling (cylinder).

The effects of laser processing on the microstructure of the QT + IH + X and the FP + DR + X process chains as a pure thermal process are shown in Figure 14. A hardened layer is resulting from these processes. Differences in the morphology and size of the microstructural constituents can be observed, in particular for the QT + IH + LM. The surface layer consists for all samples of a fine granular martensitic structure. In the laser melted sample surface almost structure less areas are visible. In the former ferritic-perlitic samples, a fine grid of cementite lamellae is still visible.

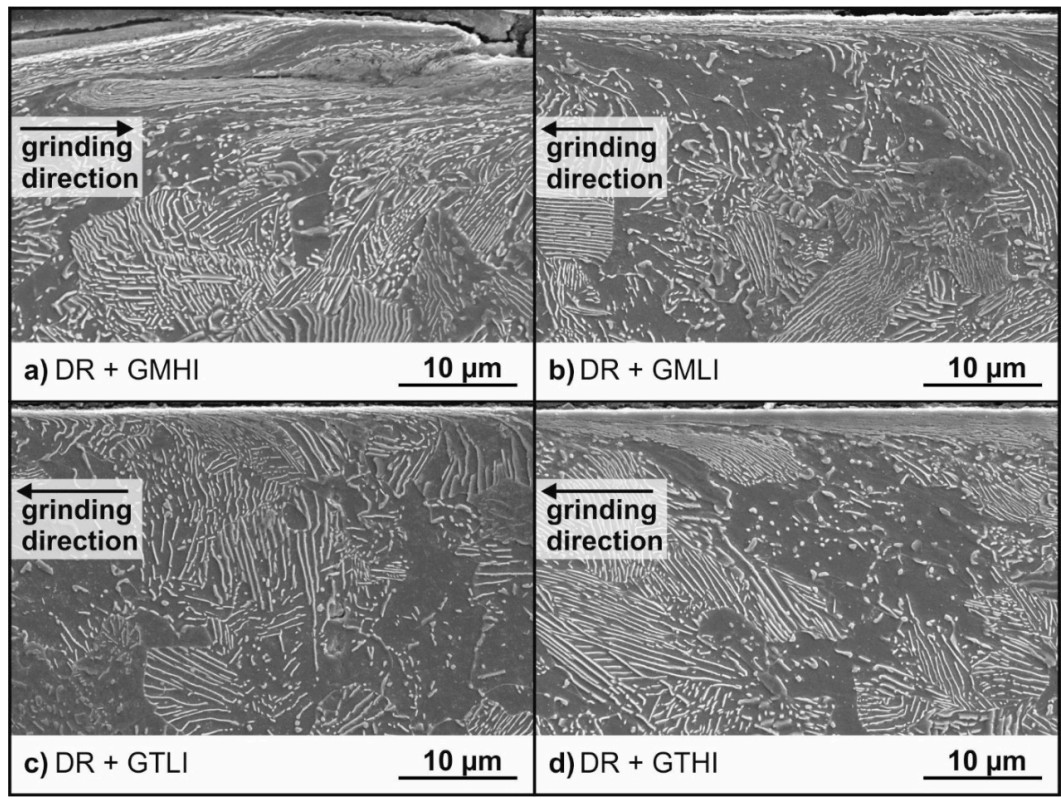

**Figure 13.** Near surface microstructure of process chains consisting of FP + DR +: (**a**) GMHI, (**b**) GMLI, (**c**) GTLI, and (**d**) GTHI (cylinder).

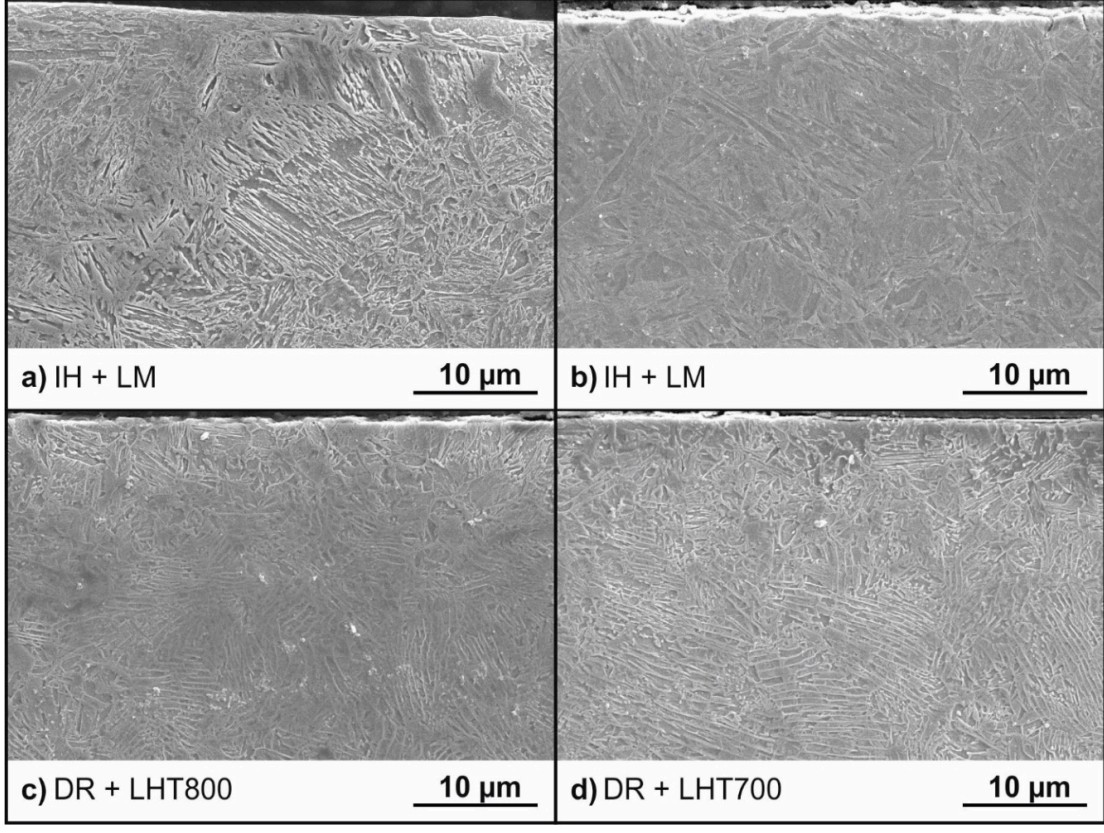

**Figure 14.** Microstructure of the samples processed with (**a**) laser melting and (**b**–**d**) laser hardening treatments.

On the contrary, the initial microstructure of EDM-treated samples is completely preserved beneath a thin rim of about 0.5 µm to 1 µm where the microstructure is strongly modified. Figure 15 shows exemplarily the ferritic-perlitic and deep rolled state after EDM finishing treatment. Due to the spark ruled process melt drops of different sizes stick on the surface.

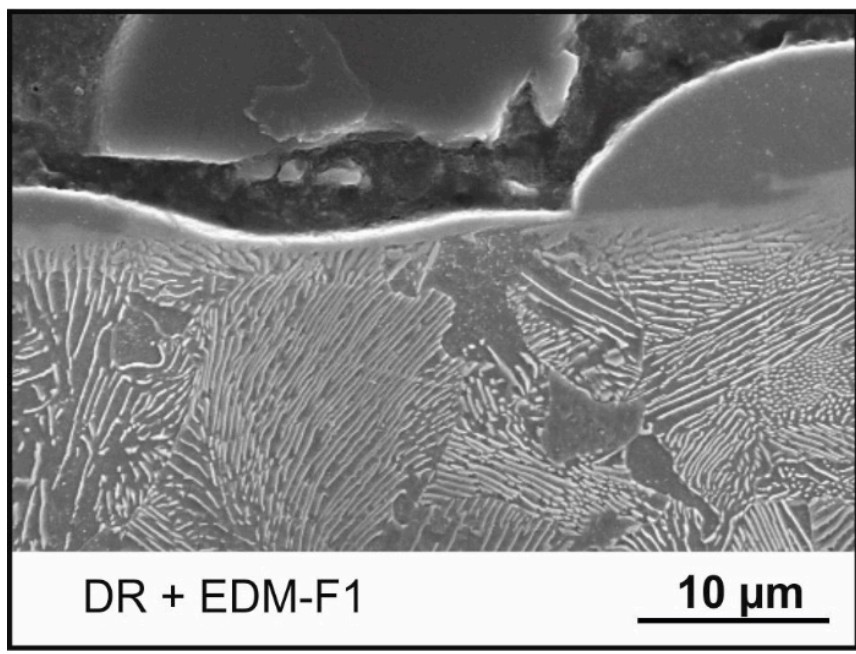

**Figure 15.** Microstructure of FP deep rolled sample processed with EDMF1 process (cuboid).

### 3.2.2. Investigations Using Electron Backscatter Diffraction

In the following section, results of EBSD analyses will be presented as a function of the distance from the initial surface. Figures 16 and 17 show the grain sizes and kernel average misorientation of the cross sections of the surface zones after QT + IH + grinding, and laser processing, respectively. Prior induction hardening caused a phase transformation within the first 800 µm (cylinder sample), resulting in a reduced and homogeneous grain size.

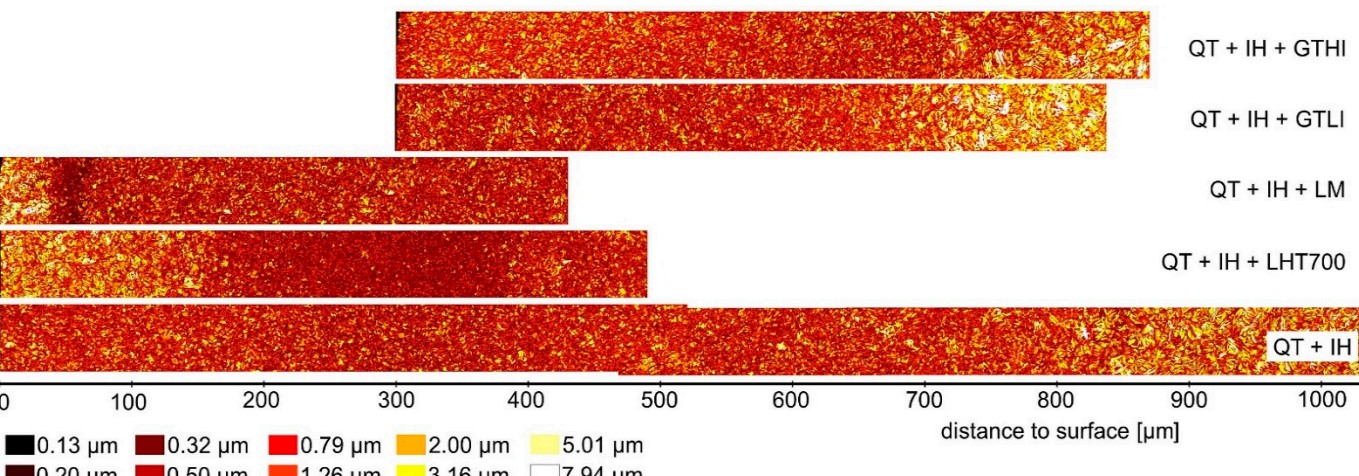

**Figure 16.** EBSD grain size maps of QT, inductive hardened, grinding and laser processing, respectively.

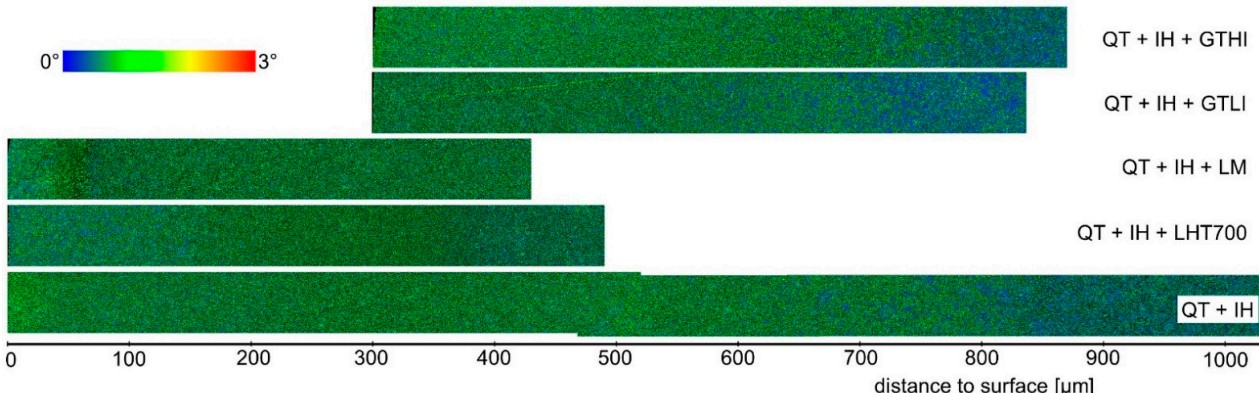

**Figure 17.** EBSD kernel average misorientation map (1st neighbor, 5°, perimeter) QT, IH, grinding and laser processing, respectively.

Figure 18 summarizes the average grain size and KAM values for the different processes as a function of the depth from the initial surface from the previous maps. The influence of the different processes can be observed and assessed in a more quantitative manner, and the already described trends can be confirmed. Additional phase transformation due to following laser processing results in even smaller grain sizes in a distance of 50 µm to 200 µm for LM and in a distance of 150 µm to 400 µm for LHT700 (cf. Figure 18a)). Closer to the surface, the high temperatures during laser processing cause a grain growth, resulting in larger grain sizes than the former induction hardening state. Since grinding with thermal impact parameters did not result in a temperature increase above the austenitization temperature, the IH initial state is tempered. For the sample with high thermal impact (GTHI), an increase in grain size can be observed. Since no phase transformation took place, thermally activated recovery mechanisms occurred, leading to stress relaxation and a reduction in the number of LAGBs resulting in larger grain sizes. For the sample with small thermal impact (GTLI), the temperature generated during grinding is too low to observe clear modification of the microstructure. The small drop in grain size for GTHI at 300 µm—which is actually at the surface of GTHI—is attributed to the mechanical impact and grain deformation at the processed surface. The KAM values in Figure 18b correspond reciprocally to the grain sizes, as smaller grains formed by phase transformation or massive plastic deformation usually have a more distorted crystal structurea and therefore higher KAM values.

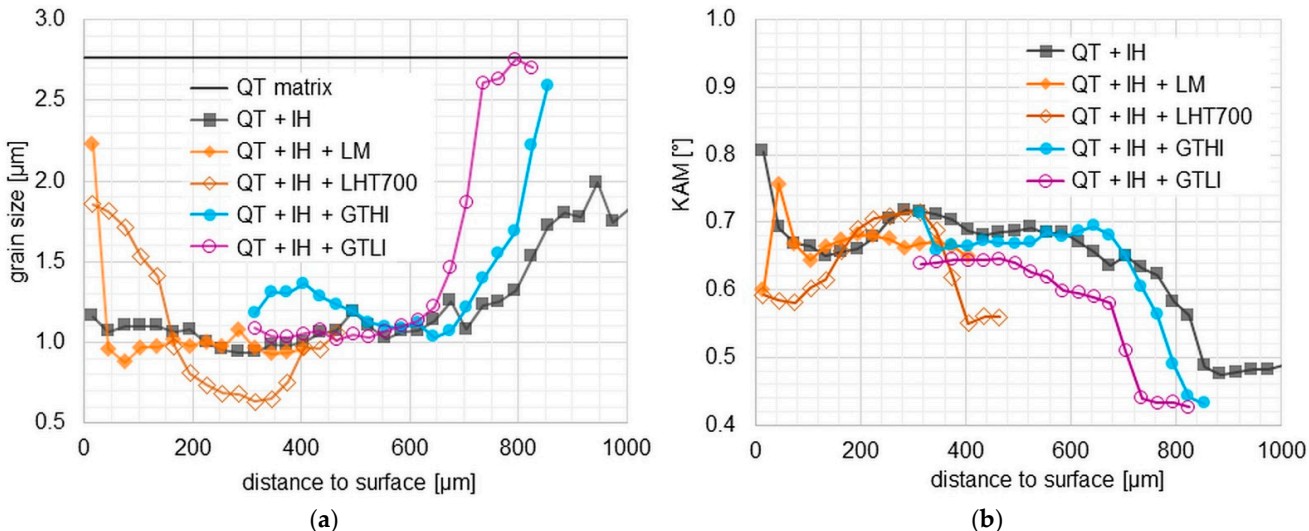

**Figure 18.** Comparison of (**a**) measured grain sizes and (**b**) kernel average misorientation after laser processing and grinding of induction hardened cylinders with two parameter sets.

Figure 19 shows the image quality with colored grain boundaries of the samples with FP + DR pre-treatment. In Figures 20–22 the average data corresponding to these maps are presented for quantitative evaluation. Deep rolling caused formation and rearrangement of dislocations, forming dislocation networks and sub-grains within the former ferrite-perlite grains due to dynamic recovery. This is visible in the increase in the grain boundary length at the surface of FP + DR for LAGBs (1–5° and 5–15° misorientation) but no increase in grain boundary length for HAGBs (15–180°) as those remain the same during recovery. On the other hand, for additional processes that cause phase transformation (EDM-F1, LHT800) or dynamic recrystallization (GMHI, GMLI, PT1S, PT2S) an increase in HAGBs can be observed (Figure 22).

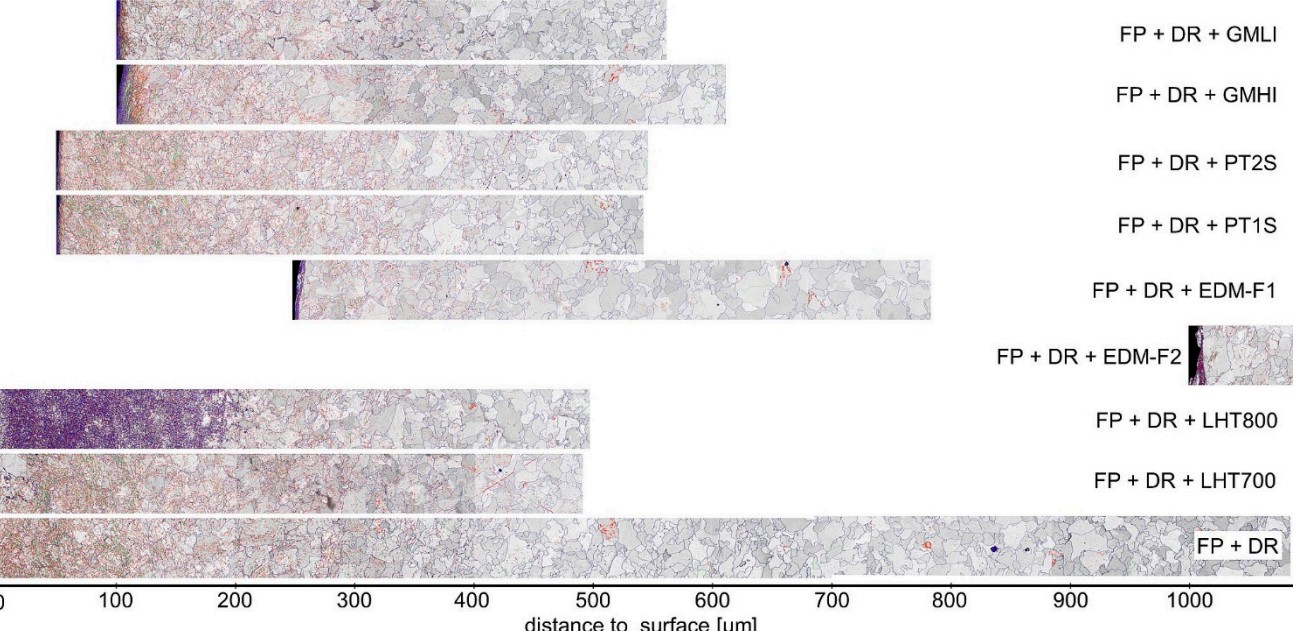

**Figure 19.** EBSD image quality maps with colored grain boundaries: HAGB in blue (15–180°) LAGBs in green (5–15°) and red (1–5°).

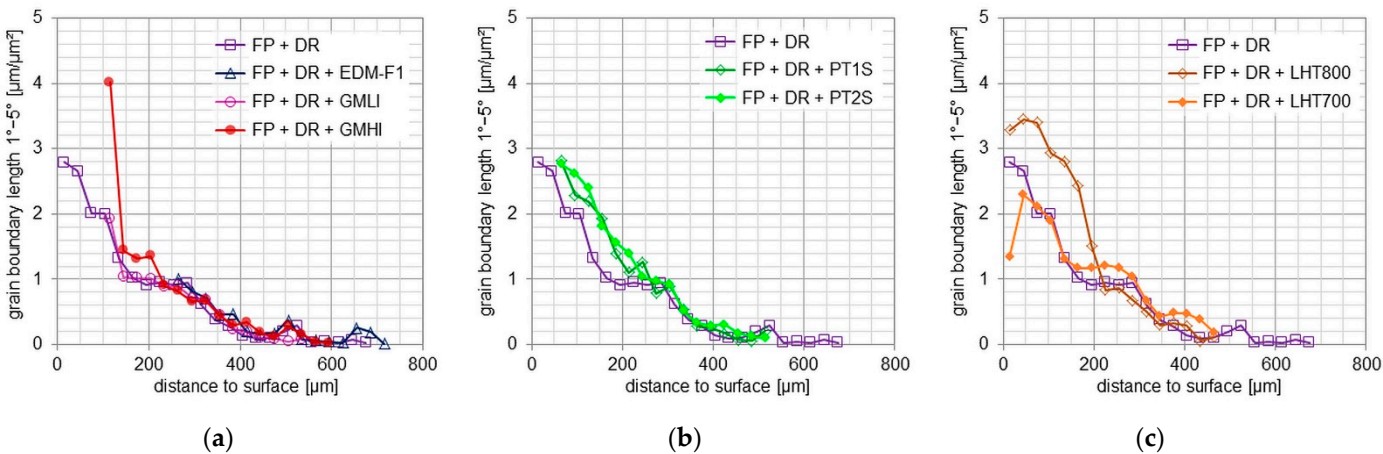

(**a**)             (**b**)             (**c**)

**Figure 20.** LAGB (1–5°) grain boundary length of FP + deep rolled +: (**a**) grinding and EDM, respectively, (**b**) turning and (**c**) laser processing samples.

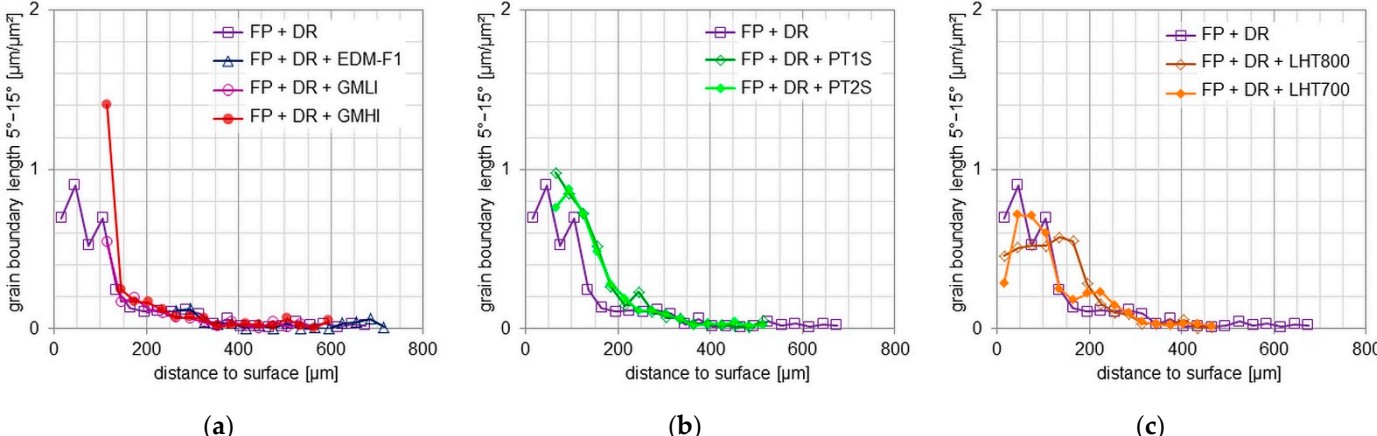

**Figure 21.** LAGB (5–15°) grain boundary length of FP + deep rolled +: (**a**) grinding and EDM, respectively, (**b**) turning and (**c**) laser processing samples.

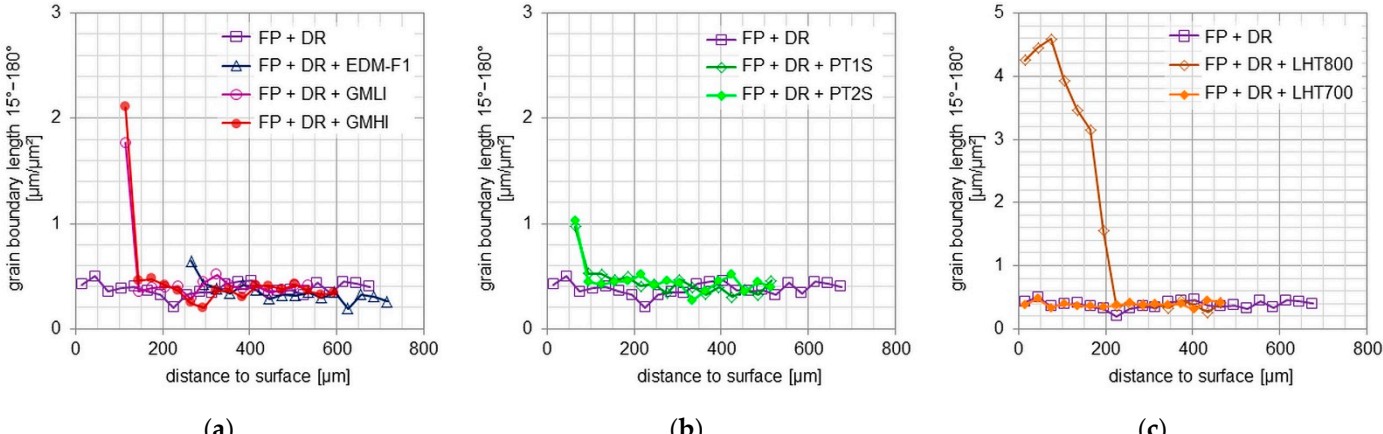

**Figure 22.** LAGB (15–180°) grain boundary length of FP + deep rolled +: (**a**) grinding and EDM, respectively, (**b**) turning and (**c**) laser processing samples.

For grinding with thermal impact (GMHI), precision turning (PT1S, PT2S), and laser processing (LHT800) sub-grain boundaries are formed in the surface zone (Figures 20 and 21), whereas for EDM-F1, no additional sub-grain boundaries are formed and for LHT700, part of the sub-grain-boundaries and dislocation networks from the former deep rolling are removed.

Figure 23 shows the kernel average misorientation maps of samples with FP + DR pre-treatment. In Figure 24 the average data corresponding to the maps is presented for quantitative evaluation. For grinding with thermal impact (GMHI), precision turning (PT1S, PT2S), and laser processing (LHT800) lattice distortions is occurring in the surface zone (Figures 20, 21 and 23).

The EDM specimen with a previous deep rolling process shows a typical rim zone at the surface. For DR + EDM-F1 with a removal of 250 μm, the EBSD KAM map shows preserved misorientation of the deep roll process. These relics cannot be found in measurement of DR + EDM-F2, which is explained by the deeper layer removal of 1000 μm during the EDM process that removed the entire layer previously affected by the deep rolling process.

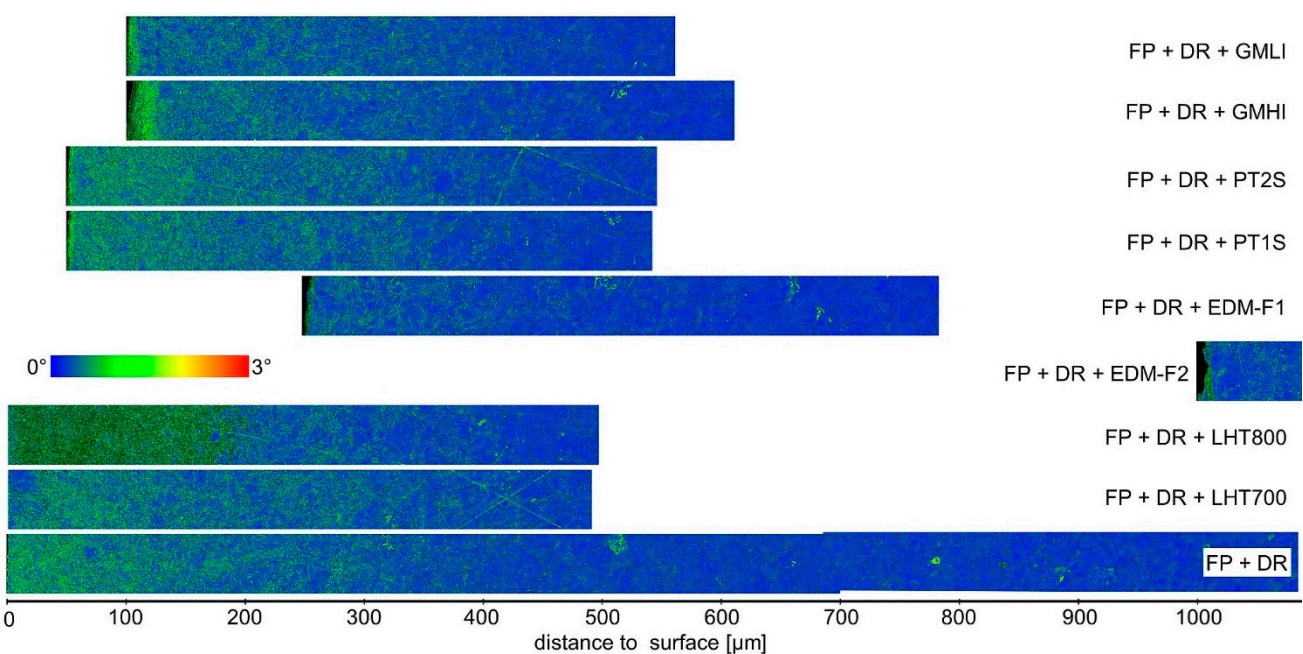

**Figure 23.** EBSD kernel average misorientation map (1st neighbor, 5°, perimeter).

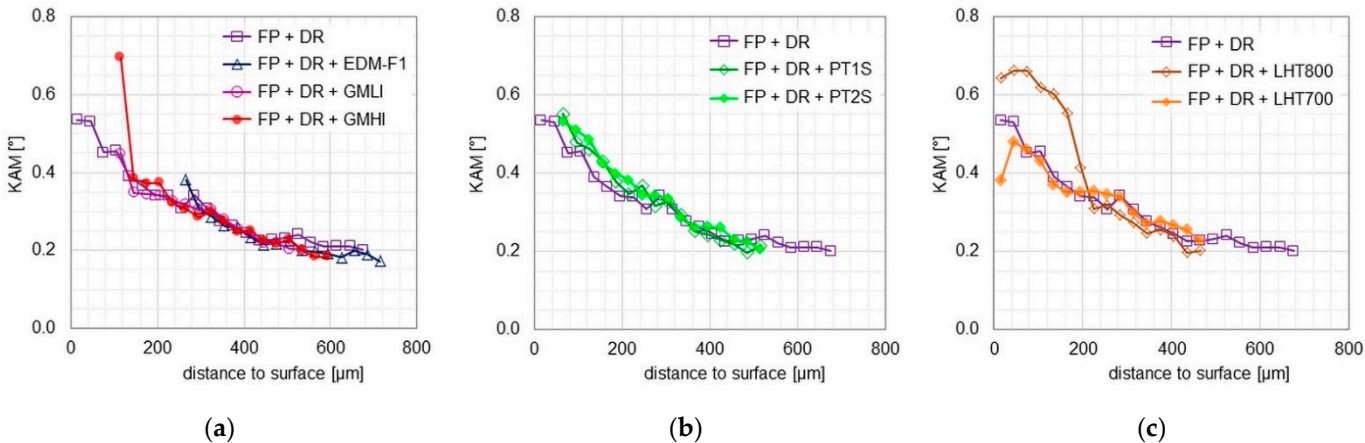

(**a**)                                       (**b**)                                       (**c**)

**Figure 24.** EBSD kernel average misorientation (1st neighbor, 5°, perimeter) of FP + deep rolled +: (**a**) grinding and EDM, respectively, (**b**) turning and (**c**) laser processing samples.

### 3.3. Comparison of XRD-Measurement Results

### 3.3.1. Residual Stresses

The initial material states exhibit an initial compressive residual stress state over large depths as starting point for all further applied processes. ECM processes show no influence on the measured residual stresses and unchanged values compared to the initial state (Figure 25). EDM leads to a strong change toward tensile residual stresses in the outmost surface layer. Since the lattice parameters measured with XRD-methods are known to tend to scatter comparatively strongly, the values measured on two cylinders are displayed in Figure 26 for the initial state, namely the induction hardened material. Further Figure 26 shows that the GMHI/GMLI processes only slightly modified the residual stresses. Both have comparably small influence zones (<50 μm), which leads to rapid transition to the initial material state. Compared to this, the high thermal impact in laser processes as well as grinding with thermal impact (GT) lead, because of the phase transformations, to much deeper modifications with mainly tensile residual stresses in the surface region and need at least >300 μm to equilibrate, as shown in Figure 26. Only the laser hardening process

LHT700 leads to only slightly reduced compressive residual stresses compared to the initial QT + IH state.

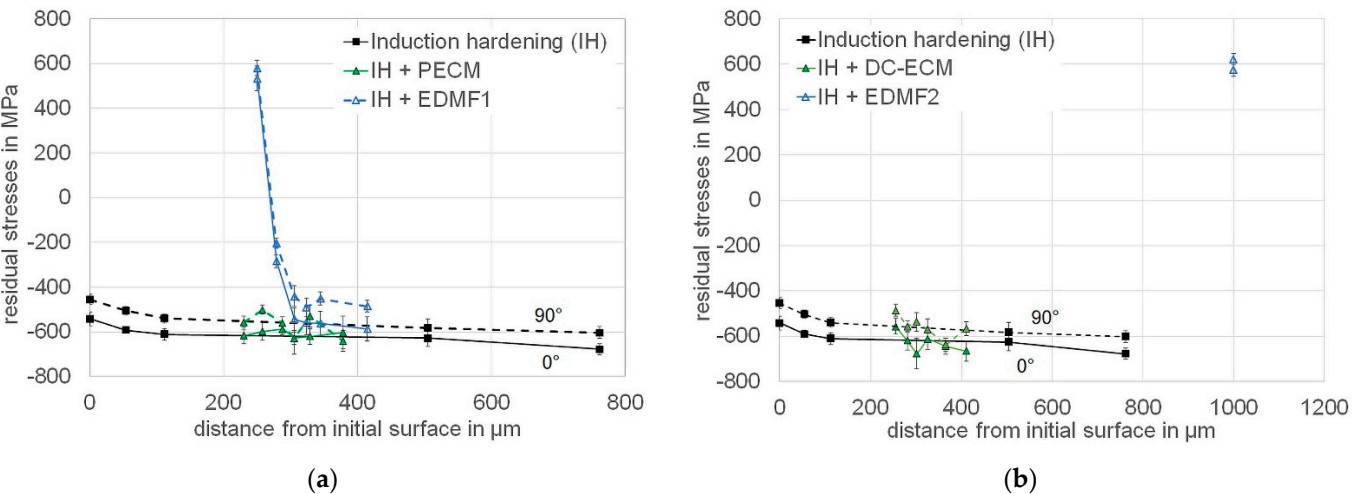

(a)

(b)

**Figure 25.** Residual stress depth profiles of induction hardened and EDM- and ECM-treated cuboids: (**a**) Parameter set 1; (**b**) Parameter set 2 (0° solid and 90° dashed lines).

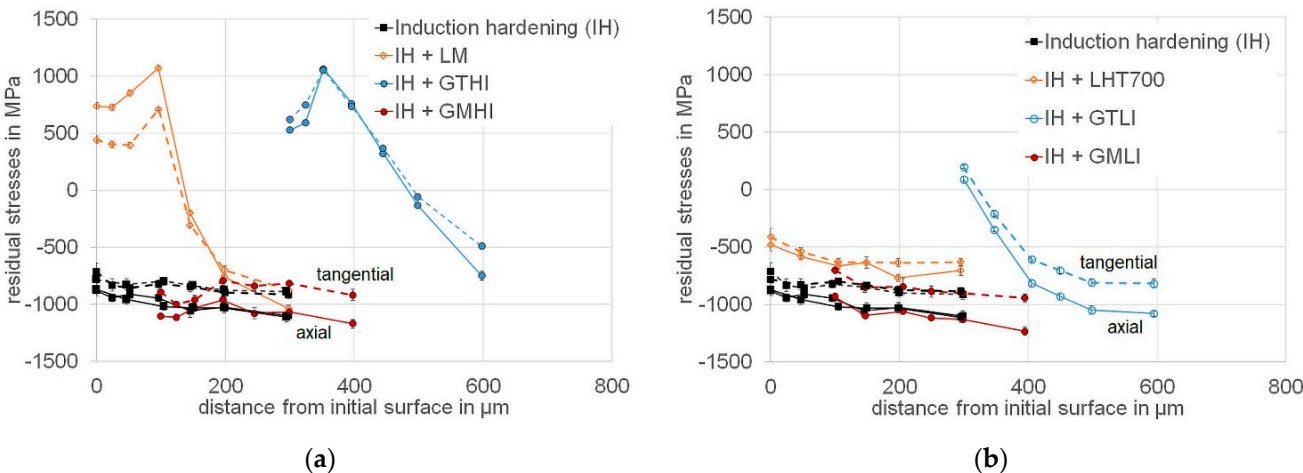

(a)

(b)

**Figure 26.** Residual stress depth profiles of induction hardened, laser-treated resp. ground cylinders: (**a**) Parameter set 1; (**b**) Parameter set 2 (solid line: axial stresses, dashed line: tangential stresses).

In the initial deep rolled state a profound compressive stress state is set up in the surface region (Figures 27 and 28). On the cuboids, the surface values in the deep rolled state were measured on different positions to show the extent of the scatter in the measured values (Figure 27). For the deep rolling process chain, no significant influence of the ECM machining (DR + DC-ECM as well as DR + PECM) on the course of the initial residual stress profile of the deep rolling process (DR) can be seen. Figure 18 shows the results for two deep rolled cylinders in the initial state. With exception of laser hardening at LHT800, all other subsequent processes tend to switch the compressive residual stress state to tensile or at least to reduce the compressive stresses slightly. LHT700 produces a martensitic surface layer and increases compressive residual stress.

Slight aberrations in the transition zone to the initial stress state can be expected, because the method of material removal leads to different depth corrections compared to the electro-polishing process used for the X-ray diffraction depth measurements. Different area, form and geometry will influence the measured residual stresses, but overall the results in depth show a good agreement for the different samples.

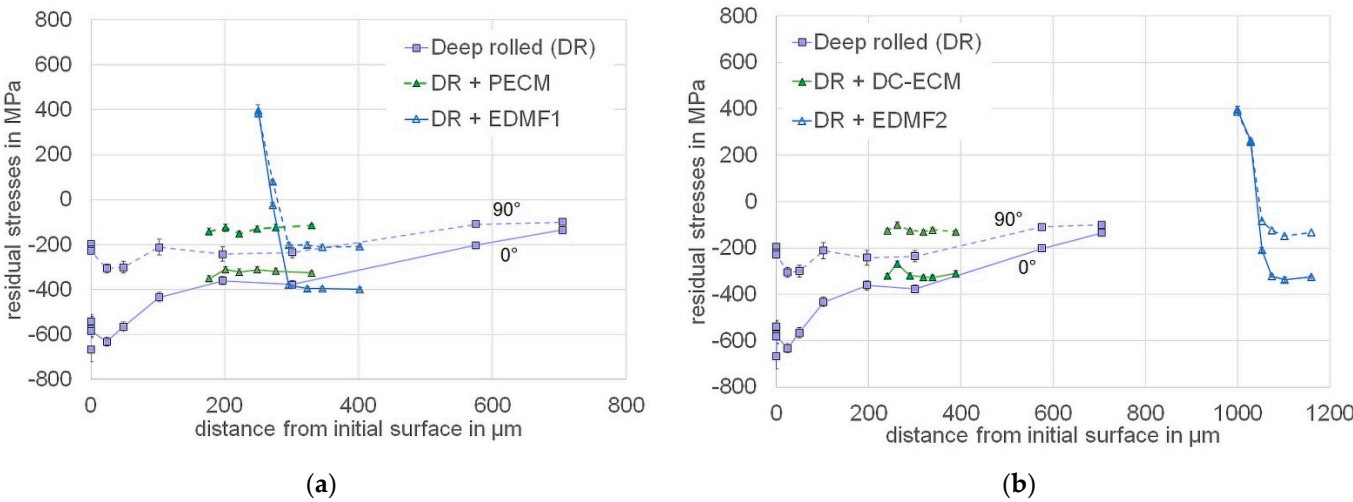

**Figure 27.** Residual stress depth profiles of deep rolled and EDM- and ECM-treated cuboids: (**a**) Parameter set 1; (**b**) Parameter set 2 (solid line: transversal to deep rolling direction, dashed line: in rolling direction).

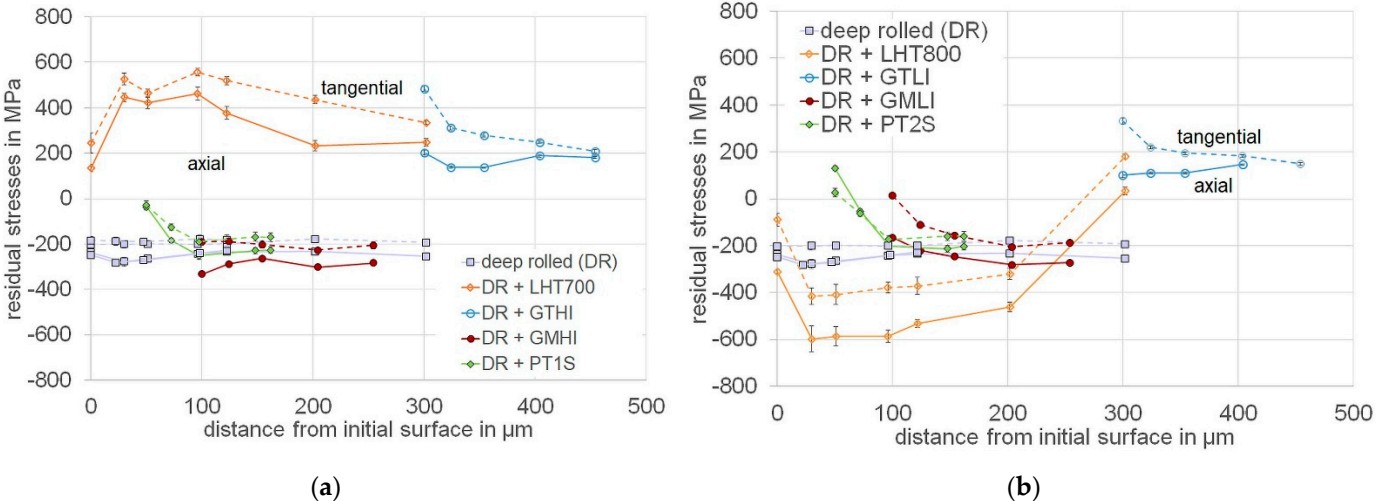

**Figure 28.** Residual stress depth profiles of deep rolled, laser-treated resp. ground or turned cylinders: (**a**) Parameter set 1; (**b**) Parameter set 2 (solid line: axial stresses, transversal to deep rolling direction, dashed line: tangential stresses, in deep rolling direction).

### 3.3.2. Full Width at Half Maximum (FWHM)

The peak width measured from the diffracted signal can give qualitative information about the depth of deformed zones or phase changes induced through thermal process mechanisms. For the induction hardened initial material state, the subsequent processes have to interact with small crystallites/grains and a high dislocation density because of the martensitic layer with initial high FWHM. As shown in Figures 29 and 30, this leads to a lowering of the local FWHM values in almost all processes in the influenced zones. For EDM, ECM, laser, and grinding processes, only dislocation annihilation or rearrangement occur, with the ECM processes again showing no significant effect on the surface material state. Only laser hardening leads to an increase in the peak width in a limited depth due to the formation of a new, martensitic layer. On the other hand, the LM process reduces FWHM drastically, which was not expected, since self-quenching and formation of a hardened layer should also have occurred. On the other hand, the thermal input from the GT grinding creates a significant decrease in the FWHM in a layer with varying depth and with varying intensity, directly related to the employed process. The affected layers align with the projected process zones from the residual stress measurements, but the

crossing point into the material state occurs at lower depths, meaning the residual stresses generated from the material state mismatch need additional material volume to equalize.

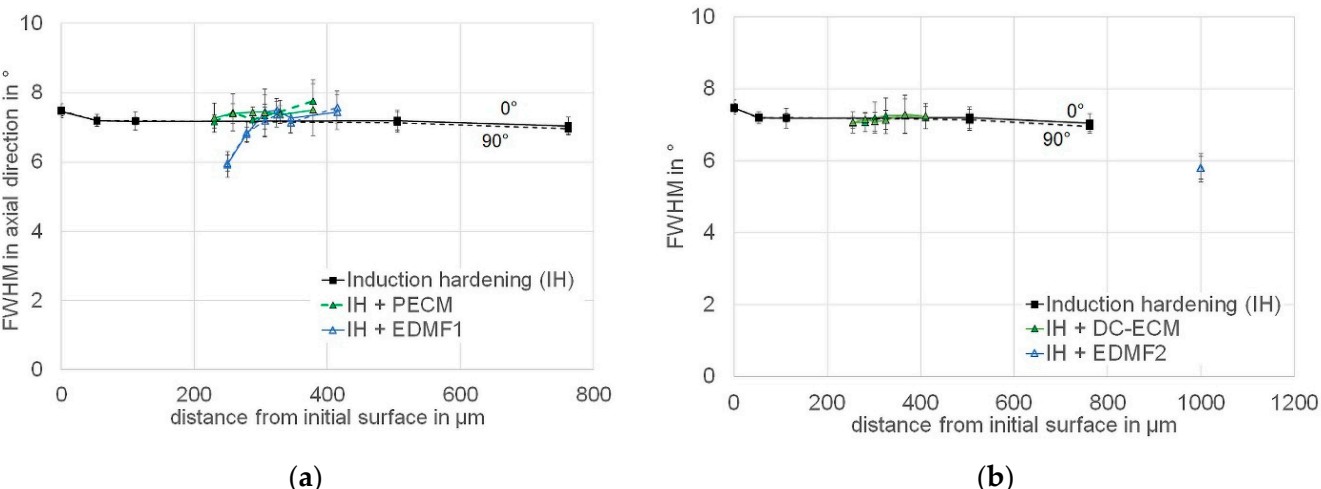

(**a**)          (**b**)

**Figure 29.** FWHM depth profiles of induction hardened and EDM- and ECM-treated cuboids: (**a**) Parameter set 1; (**b**) Parameter set 2 (0° solid and 90° dashed lines).

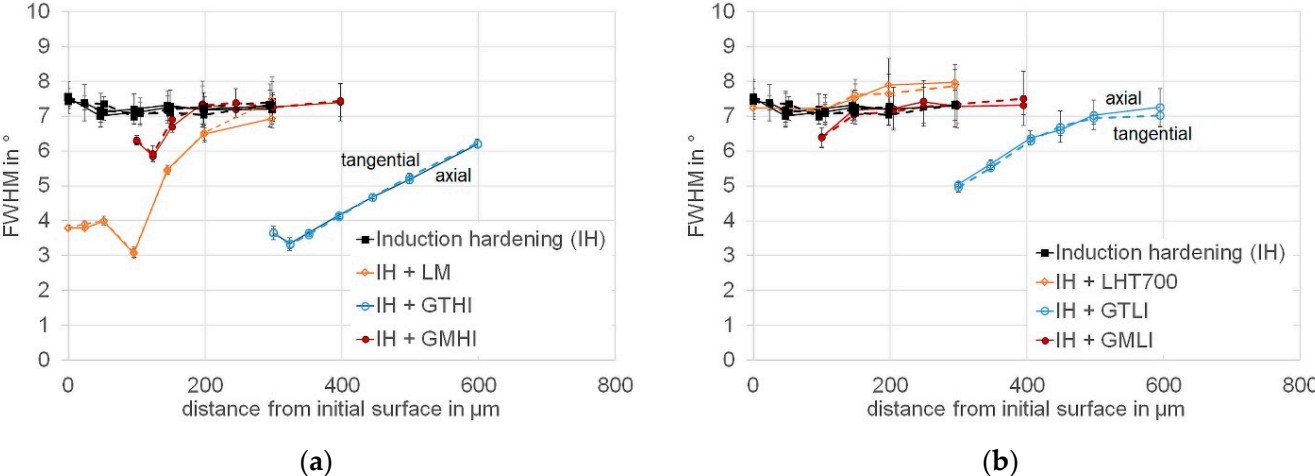

(**a**)          (**b**)

**Figure 30.** FWHM depth profiles of induction hardened and laser-treated resp. ground cylinders: (**a**) Parameter set 1; (**b**) Parameter set 2 (solid line: axial stresses, dashed line: tangential stresses).

For the deep rolled material, material under contact experienced plastic deformation above the yield stress of the material, as visualized by the continuous increase in the FWHM towards the surface, which is more pronounced in the case of the cuboid samples (Figure 31) than in the cylindrical samples (Figure 32). The observed process impacts in this study can increase FWHM due to additional thermal or mechanical effects in EDM, grinding (GM parameters), turning and laser hardening processes. For grinding (GM parameters) and EDM, a material change in the first 50 µm of the process zone is generally observed with notable increased FWHM, before the values again match to the initial material state. Increasing the thermal impact of the grinding GT process parameter set leads to noticeable modification of the microstructure; however, this is still limited to a very shallow depth. A notable exception is the laser hardening process LH800 at higher power, where the strong increase indicates a hardened martensitic layer, consistent with the microscopy results.

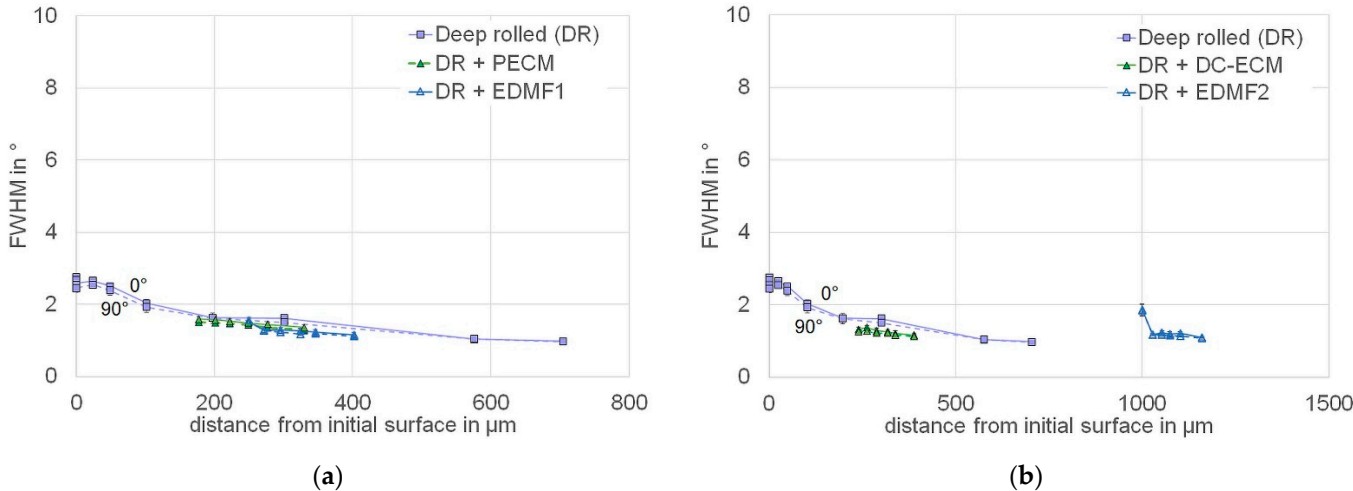

(**a**)

(**b**)

**Figure 31.** FWHM depth profiles of deep rolled and EDM- and ECM-treated cuboids: (**a**) Parameter set 1; (**b**) Parameter set 2 (solid line: transversal to deep rolling direction, dashed line: in rolling direction).

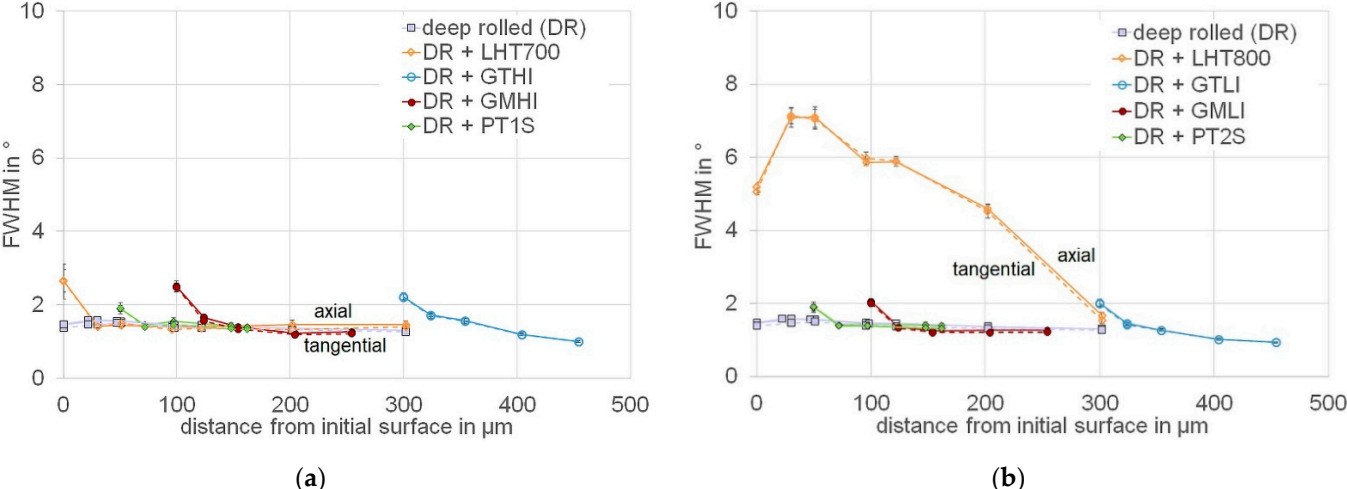

(**a**)

(**b**)

**Figure 32.** FWHM depth profiles of deep rolled, laser-treated resp. ground or turned cylinders: (**a**) Parameter set 1; (**b**) Parameter set 2 (solid line: axial stresses, transversal to deep rolling direction, dashed line: tangential stresses, in deep rolling direction).

While process zones, transitions and phase changes can be appropriately identified using the peak width data for different processes, contact conditions and mechanisms, the FWHM allows no direct correlation or prediction of the sign (tensile or compressive) and depth evolution of the residual stress state that is created.

## 4. Discussion

Based on the different results, the process impact and the effect of overlapping process modifications for the different initial conditions can be assessed. First, the different processes are discussed separately and in a second step, global assessment of the results is provided.

### 4.1. Grinding

Depending on the intensity range of thermo-mechanical loads, grinding is characterized by a broad achievable spectrum of modifications. This can be seen in the ferritic-perlitic microstructure (Figure 13) and especially in the residual stress depth curves (Figure 28).

Grinding with main mechanical impact with high intensity is able to increase the compressive residual stresses by about 100 MPa compared to the deep rolled state at a

corresponding depth (Figure 28). A reduced mechanical impact in the second parameter set increases the thermal impact and therefore causes a shift of the residual stresses towards tension by about 70 MPa. In both cases, the workpiece experiences a grain refinement as a result of work hardening with a temperature less than 300 °C. This is noticeable in the EBSD measurements (Figure 22) as well as in FWHM values (Figure 32). Furthermore, an increase in Martens hardness is visible (Figure 9). By a further increase in thermal loads (T < 600 °C), tensile residual stresses in the surface of about 100 MPa are caused by GTLI (Figure 27). The highest thermal loads (GTHI) result in tensile residual stresses of about 200 MPa at the surface. Electron microscopic investigations confirmed that thermal damage occurred. As a result of the high surface temperatures (T > 850 °C), a re-hardened zone appeared directly at the surface, also visible in the hardness measurement results (Figure 9) and the FWHM results at the surface (Figure 32).

Similar results can be obtained for the quenched, tempered and induction hardened state with the different grinding operations. The highest mechanical impact (GMHI) leads to a slight increase in compressive residual stresses, while GMLI shows no significant influence (Figure 30). With increasing thermal impact (GTLI, GTHI) the modifications are shifted towards tensile residual stresses up to 1000 MPa in the case of GTHI. This modification due to a strong tempering zone is reduced to a small surface area (ca. 200 μm, see Figure 26a). The hardness results show a reduction in the area beneath this zone (Figure 8). The EBSD measurements show that the thermal impact results in a slight grain growth in the near surface area (Figure 18).

### 4.2. Precision Turning

The results measured on the precision turned workpieces (Figures 9, 20–22, 24, 27 and 32) correspond to other investigations, where the material modifications as well as the thermo-mechanical material loads were determined experimentally and numerically [9,10]. The depth impact of precision machining remains comparatively low. The variation in the process in this investigation consisted of one specimen machined in one stage (PT1S: $a_p$ = 50 μm), while the other specimen was machined in two stages (PT2S: $a_{p,1}$ = 30 μm, $a_{p,2}$ = 20 μm). EBSD images (Figures 19 and 23) of the machined specimen show a similar material modification for both PT1S and PT2S, as the resolution and scale are not sufficient for the small influenced surface and subsurface zone with a maximum depth of 10 μm. Still, a distinct increase in HAGBs (Figure 22) at the surface confirms the formation of nanocrystalline grains. The hardness measurements (Figure 9) and FWHM depth profile (Figure 32) also allow the detection of differences from the deep rolled material only at the very surface. On the other hand, the depth profiles of residual stresses (Figure 27) show a different value for PT1S and PT2S near the surface, with a higher modification towards tensile stresses in PT2S indicating a higher thermal material load, than with PT1S. Compared to the other machining processes, precision turning only modifies the material in the nearest subsurface zone (<10 μm) and does not affect underlying material modifications from previous processes (i.e., deep rolling) with a higher depth impact.

### 4.3. Laser Hardening and Melting

The high thermal energy input allows the precise control of both the phase and the depth of the process zone (cf. Table 8). For the deep rolled material with its ferrite-perlite basic structure, the process with a laser power of 700 W (surface temperature 940 °C) changes the compressive residual stresses (cf. Figure 28) to tensile stresses. Because of the high $A_{C1}$ temperature [11], the phase transformation to austenite is incomplete. Figures 9, 14, 19, 23 and 32 confirm this interpretation. The thermal effect dominates the results and leads to tensile stresses in the surface [12]. The slightly higher laser power of 800 W achieved a surface temperature of 1250 °C and leads to a full austenitization within the surface near area (approximately 200 μm). In that case, approximately −600 MPa pressure was achieved (cf. Figure 28b). The formation of austenite and subsequent hardening

commonly leads to compressive stresses within the hardened zone and changes to tensile stresses when the not hardened area is reached (cf. Figure 28b) [15].

In case of induction hardened QT material the laser power of 700 W is sufficient for a complete austenitization. The $A_{C1}$ temperature is about 80 K lower than for the ferrite/perlite material [11], since the surface of the induction hardened sample is darker than the mirror blank deep rolled surface. Hence, the absorption rate is higher and leads therefore to a surface temperature of about 1250 °C (cf. Table 8). The Martens hardness (cf. Figure 8b) in combination with the results of FWHM measurements (cf. Figure 30b) confirms the formation of a new hardened zone. The compressive residual stresses decrease significantly from about −800 to −500 MPa. The reason for this decrease is not completely clear. Similar experiments without previous induction hardening show similar residual stresses within the hardened zone [15].

The laser melting induced a strong decrease in hardness and FWHM while inducing high tensile residual stresses at the near surface, which was not expected at first, since a sre-hardening upon cooling took place here as well. However, due to the very high temperature and the long processing time, grain growth occurred and lattice distortions recovered (see Figure 18), which explains the modified final properties. Additionally, strong decarburization of the remelted surface boundary layer can occur, which also reduces hardness and FWHM [22].

*4.4. EDM*

The measured residual stresses at specimens IH + EDMF1 and IH + EDMF2 are about $\sigma_{surface}$ = 600 MPa and drop within about $\Delta z$ = 50 µm (Figure 25). This is in good agreement with further analysis also showing no significant influence of the ECM process on a pre-grinding or -EDM operation, cf. [13].

The residual stress at the surface of the DR + EDMF1 as well as the DR + EDMF2 specimen reached the yield strength of the machined material in a ferritic-perlitic condition (YSFP ≈ 380 MPa) (Figure 26). In deeper regions of specimen DR + EDMF1, the measured stress drops within 50 µm to the values of the initial state in the corresponding depth. Furthermore, it is noticeable that the measurements for depths z > 300 µm have a directional dependence, which is not typical for the EDM specimen because of the stochastic character of the process. This directional dependence can also be observed for DR + EDMF2 although the EBSD images (Figure 23) and the FWHM values (Figure 31) indicate that the whole impact of DR should have been removed.

As an example for the limited thermal impact close to the surface, the impact of EDM finishing processes in the FP-annealed state are compared to samples with and without deep rolling treatment (Figure 33). The comparison shows that the change in the residual stress state due to EDM can almost be superimposed to the initial state in a multi-step process, but the removal due to the process has to be taken into account. In a similar way to the residual stresses dropping to the initial state after a few µm in the FP-annealed state (<40 MPa), they fall to the initial residual stress state in the deep rolled state. While the EDM machined surface shows no preferred orientation for the stresses, the values for the superimposed states begin to split up in depth to achieve a smooth transition to the deep rolled state with its lower transversal (perpendicular to machining direction) and higher axial (machining direction) stress orientations.

*4.5. ECM*

Both the PECM and DC-ECM process have shown to conserve to original residual stress state without further material modification in depth outside its specific surface layer. While for stress superposition and surface alteration, this is not an optimal result, on the other hand, this means that geometry changes may be achieved through this process while preserving the initial process state, if the layer removal does not completely remove the modified zone. This, however, might be compensated by increasing the depth of the previous process step. More detailed analysis in future should focus on the detailed

correlation between the residual stress profiles of different graded ECM and EDM processes and the resulting part functionality, cf. [14].

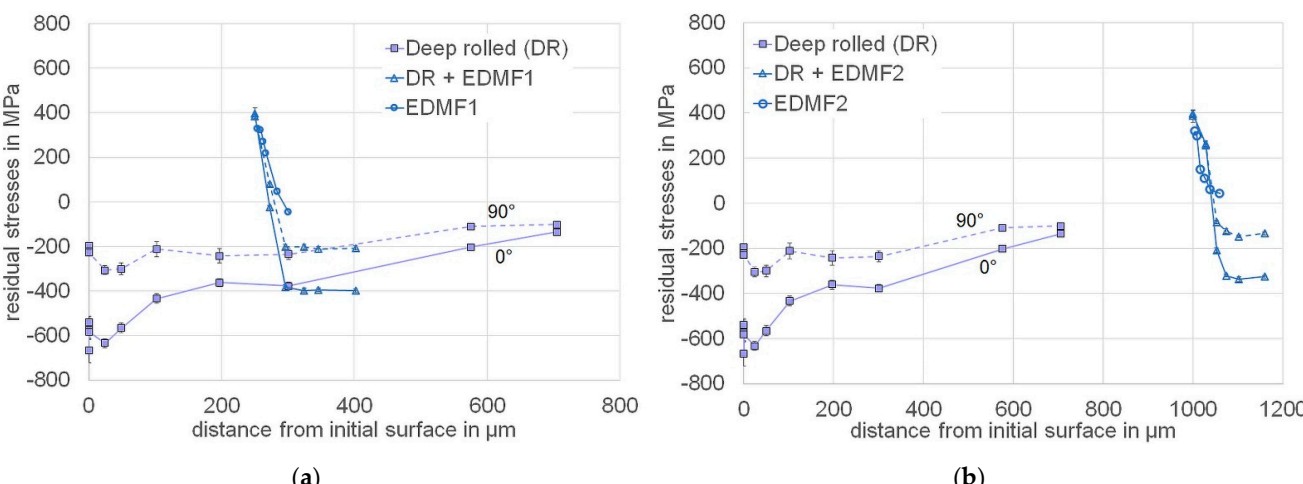

(**a**)

(**b**)

**Figure 33.** Residual stress depth profiles of FP-annealed deep rolled, deep rolled and EDM-treated, and just EDM-treated cuboids in two different finishing parameter sets: (**a**) Parameter set 1; (**b**) Parameter set 2 (solid line: transversal to deep rolling direction, dashed line: in rolling direction).

### 4.6. Overall Process Comparison, Reliability of Measurement Results, and Influence of the Initial State

Since chemical composition and therefore the properties of natural materials cannot be assumed to be homogeneous over several samples, close attention must be paid to the spread of the measured values.

Figure 8 shows the hardness depth profiles of the initial induction hardened cylinders. The spread from cylinder to cylinder can be estimated by the differences in the basic values, towards which the measured values of additionally manufactured surfaces are striving. The strong scatter within the measured values of one depth profile can be contributed to the measurements inaccuracy due to the very small indentation of 10 mN. On this basis, the depth profile of the initial state can be assumed not to be influenced by a main mechanical impact. The processes with low thermal impact have no influence on the near surface microstructure (Figure 10), the grain size and the kernel average misorientation measured by electron microscopy (Figure 18). A thermal impact due to GT processes led to thermally activated recovery mechanisms and therefore a reduction in hardness and a reduction in LAGBs (Figure 18). Further enhanced thermal impact leads to re-austenitizing and re-hardening of the surface, resulting in the initial hardness (Figure 8). Direct changes in the microstructure due to the laser melting process are only detectable in the outer fifty microns with an increase in grain size and a decrease in the KAM-values (Figure 18). This corresponds with a decrease in hardness (Figure 8). The further decrease in hardness in greater depths compared to the initial state can be attributed to a tempering of the induction hardened microstructure.

Figure 9 shows the hardness depth profiles for the annealed and deep rolled state. Due to the ferritic-perlitic microstructure (Figures 12 and 13) the spread in the Martens hardness values is even higher than in the QT state. The small indentations are influenced even by local differences in the microstructure. This behavior can also be seen in the increased standard deviation of individual depth values. The spread from cylinder to cylinder, which can be estimated by the differences in the basic values, towards which the measured values of additionally manufactured surfaces, is comparatively small. With the parameters used in this investigation, a hardness increase in the surface region was reached by deep rolling (Figure 9). Processes with main mechanical impact (PT and GM) induce a slight hardness reduction in the outer surface. Since an increase in the number of high

angle grain boundaries can be measured in the same region (Figure 22), presumably the additional mechanical load led to a dynamic recrystallization due to higher movability of dislocations in the highly deformed surface layer. An increased mechanical impact, such as that which occurs due to GMHI, causes a rise in the hardness within the first 20 μm of the surface layer. The near surface microstructure (Figure 13) shows an extreme additional plastic deformation of the surface, and in Figure 19, a strong increase in the low angel grain boundaries is displayed. The kernel average misorientation also reaches a peak value for this process combination. In combination, these results speak for an increased dislocation density and therewith a work hardening of the material. With enhanced thermal impact, grinding leads to a slight further hardness increase in the outer surface and a slight decrease in the further course of depth (Figure 9). The near surface microstructure explains this with a thin fringe of re-hardened microstructure and deformed cementite lamellae (Figure 13). The biggest changes in hardness are due to the laser processes (Figure 9). Complete austenitization is not yet achieved with LHT700, resulting in a scatter afflicted hardness profile. LHT800 leads to a complete re-hardening of the surface with a hardness depth of more than 100 μm. Though the hardness is increased with LHT700, the process, in contrast to LHT800, does not affect the high angle grain boundary lengths (Figure 22c). The difference in the surface temperature (940 compared to 1240 °C) despite the very short duration of thermal impact provoked a difference in grain growth behavior.

Since lattice parameters are particularly susceptible to differences in structure and chemical composition, two cylinders were measured in the initial state. In Figures 26, 28, 30 and 32, it can be seen that the differences between the two cylinders treated with the same parameters are negligible compared to the changes that occur due to the different manufacturing processes. Figure 34 shows a summary of the axial surface residual stresses and the maximal measured residual stress values in comparison to all investigated material states. The results are arranged from machining processes with main mechanical impact to thermal to main chemical impact.

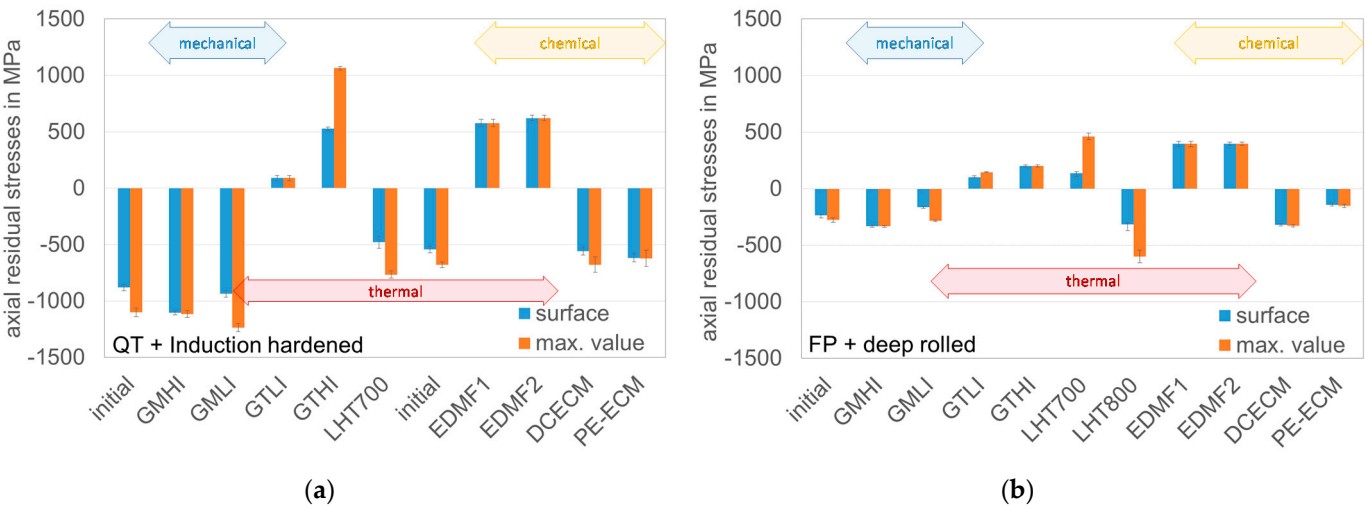

**Figure 34.** Comparison of surface and maximal residual stresses in the different investigated states: (**a**) QT + induction hardened; (**b**) FP + deep rolled.

It can be observed that the processes with mechanical impact do not strongly modify the initially present residual stress state, or enhance the compressive state. This is valid for both initial microstructures QT + IH and FP + DR. When thermal impact increases, increasing modification takes place and increasingly tensile residual stresses are generated, unless (re-)austenitizing temperature is reached during the process, generally causing compression again. Chemical impact during the processes did not significantly affect the initial stress state apart from removal.

Comparing the two different series of initial states, it can be clearly observed that in any case, the residual stress level which can be achieved by any of the subsequent processes is directly related to the strength of the initial state. For QT + IH, residual stresses higher than ±1000 MPa are reached, while in the FP state, values are generally limited in a range from ±400 MPa.

This influence can be well illustrated by analyzing the ground material states, as shown in Figure 35. It displays the axial residual stress depth profiles of the grinding processes with high impact together with the initial states and the yield strengths of the initial states. The comparison shows that the differences in maximum residual stress values are predetermined by the yield strength. Stresses exceeding these values are relieved through plastic deformation. This explains why higher residual stresses are found in the QT + IH samples despite of nominally equal processing parameters. Even if the initial state of the surface is completely extinguished, the higher strength of the core is able to keep the balance with higher residual stresses in the surface near area.

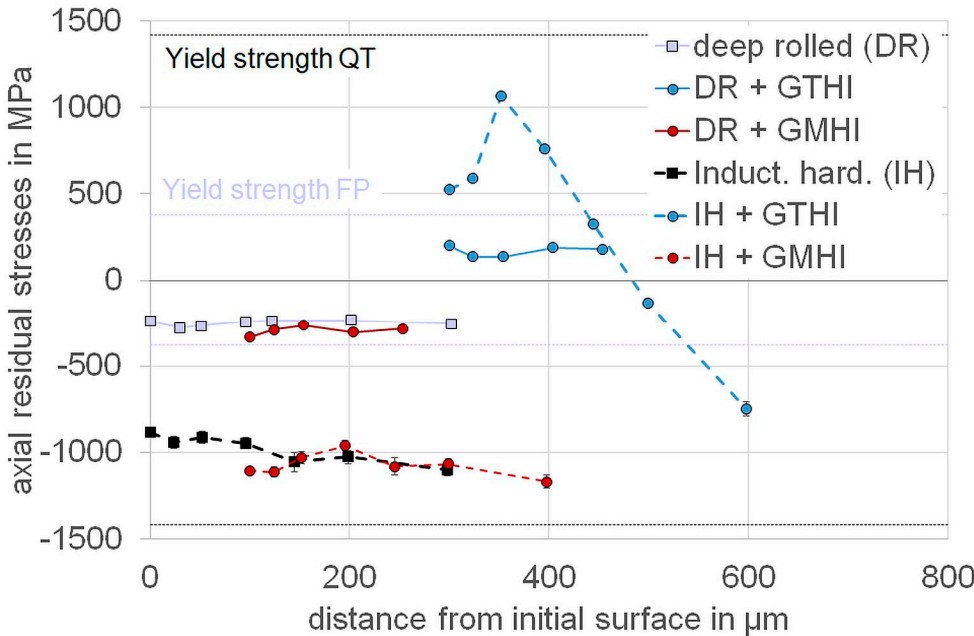

**Figure 35.** Axial residual stress depth profiles of FP-annealed, deep rolled, QT and induction hardened cylindrical samples after grinding with different thermal impact compared to the initial yield strength of the steel.

## 5. Conclusions

The results gained from this investigation are a great step towards understanding the interactions of manufacturing processes within the surface layer that impact the final surface properties of manufactured samples and components. As the investigation of these interactions under these controlled and simplified conditions is only valid so far for AISI 4140, it will be possible in the future to simulate the behavior and finally solve the inverse problem of manufacturing technologies.

The main conclusions that can be excerpted from the discussion can be separated into those that are more elementary and those that are superior. Easy to comprehend is the influence of the hardness of the initial state on the possible impact of mechanical dominated treatments, since a deformation of a harder microstructure efforts more force. On the other hand, if the enhancement in hardness is based on a work-hardening mechanism, further deformation can lead to softening due to greater mobility of dislocations.

Machining processes with main thermal impact induce mainly tensile stresses or stress relief due to annealing processes. The result of such annealing effects is dependent on the thermodynamic stability of the initial microstructure. A hardened surface will soften due

to tempering effects. A high dislocation density can be reduced due to relief mechanism. A critical degree of deformation can lead to recrystallization and therewith to the formation of a completely new micro structure due to the formation and migration of large-angle grain boundaries.

If the austenitizing temperature is reached (depending on the heating rate) and the surface is cooled sufficiently quickly, a re-hardening process can cause compressive stresses.

All these mechanisms have in common that the maximum values of resulting residual stresses is limited by the yield strength of the initial material.

**Author Contributions:** Conceptualization, B.C. and R.S.; investigation, F.B., L.C.E., M.E., F.F., M.H., E.K., Y.L., H.M., B.R., S.S., R.S. and T.Z.; project administration, B.C., J.E. and A.K.; validation, F.B., L.C.E., M.E., J.E., F.F., M.H., E.K., Y.L., H.M., B.R., S.S. and R.S.; writing—original draft, F.B., L.C.E., M.E., F.F., M.H., Y.L., H.M., B.R., S.S., R.S. and T.Z.; writing—review and editing, B.C., J.E. and A.K. All authors have read and agreed to the published version of the manuscript.

**Funding:** This research was funded by German Research Foundation (DFG) with the project number 223500200–TRR 136.

**Data Availability Statement:** Data presented in this study are available on request from the corresponding author.

**Acknowledgments:** The authors thank the German Research Foundation (DFG) for funding the transregional Collaborative Research Center "Process Signatures" with the project number 223500200–TRR 136 (Aachen, Bremen, Oklahoma).

**Conflicts of Interest:** The authors declare no conflict of interest.

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
