# Peer review of "The Influence of Former Process Steps on Changes in Hardness, Lattice and Micro Structure of AISI 4140 Due to Manufacturing Processes"

_metals, doi:10.3390/met11071102_

Round 1
Reviewer 1 Report
The presented results are highly valuable. The suggested improvements would promote the readability of the paper and the application of these results. See the comments in the attached/uploaded pdf file.

Author Response
Dear reviewer 1,
please find attached our reponse to your review. The changes due to your kind revision are marked yellow in the actual uploaded 2nd submission.
Sincerely,
Brigitte Clausen

Reviewer 2 Report
Authors presented very wide scope of investigation with appropriate experimental and test methods. Results are discussed and analyzed on the basis of measured values and adequate conclusions are given. Hence, I recommend this paper for publication with minor issues to be resolved as follows:
Line 29:
42CrMo is classified according EN 10083-3, EN 10132-2, EN 10297-1 and W.Nr. is 1.7225. Iti s recommended to use this kind of material designation and classification rather than phrase „German steel grade“.
Line 98: „First, half of the samples were quenched and tempered to 45 HRC (QT).“ There is no data about heating and cooling regime which is necessary for repeatibility of experiment. Is it possible to insert heat treatment regime parameters?
Line 162: In Table 4, values are missing for cyilnder (Maximal generator current, Frequency)!? Please replace the words cylinder value with adequate current and frequency.
Author Response
Dear reviewer 2,
please find attached our reponse to your review. The changes due to your kind revision are marked green in the actual uploaded 2nd submission.
Sincerely,
Brigitte Clausen

Reviewer 3 Report
I suggest to change the Conclusions sections from a bulleted form to a descriptive form. Also, to add some future directions therein. I also suggest to rewrite introduction to be less hastily written, i.e., expressions "In [5] ....." should at least be exchanged by "In Ref. [5] the Authors.......". I also did not see a real purpose and future applications of this study. These should be inserted, since for a moment it is more of research report than a scientific paper.
Author Response
Dear reviewer 3,
please find attached our reponse to your review. The changes due to your kind revision are marked blue in the actual uploaded 2nd submission.
Sincerely,
Brigitte Clausen
